# Inhibition of the UFD-1-NPL-4 complex triggers an aberrant immune response in *Caenorhabditis elegans*

**Rajneesh Rao[1], Alejandro Aballay[2], Jogender Singh[1]\***

[1]Department of Biological Sciences, Indian Institute of Science Education and Research, Mohali, India; [2]Department of Genetics, The University of Texas MD Anderson Cancer Center, Houston, United States

## eLife Assessment

In this **valuable** manuscript, Rao and colleagues investigate the UFD-1/NPL-4 complex, which is involved in extracting misfolded proteins in the plasma membrane and the accumulation of pathogenic bacteria in the intestine. Using **convincing** methods, the authors find that knockdown of the *ufd-1* and *npl-4* genes leads to shortened lifespan of the nematode *C. elegans* and reduced accumulation of the bacterial pathogen *P. aeruginosa* in the intestine.

**\*For correspondence:**
jogender@iisermohali.ac.in

**Abstract** The UFD-1 (ubiquitin fusion degradation 1)-NPL-4 (nuclear protein localization homolog 4) heterodimer is involved in extracting ubiquitinated proteins from several plasma membrane locations, including the endoplasmic reticulum. This heterodimer complex helps in the degradation of ubiquitinated proteins via the proteasome with the help of the AAA+ATPase CDC-48. While the ubiquitin-proteasome system is known to have important roles in maintaining innate immune responses, the role of the UFD-1-NPL-4 complex in regulating immunity remains elusive. In this study, we investigate the role of the UFD-1-NPL-4 complex in maintaining *Caenorhabditis elegans* innate immune responses. Inhibition of the UFD-1-NPL-4 complex activates an aberrant immune response that reduces the survival of the wild-type worms on the pathogenic bacterium *Pseudomonas aeruginosa* despite diminishing colonization of the gut with the bacterium. This aberrant immune response improves the survival of severely immunocompromised worms on pathogenic bacteria but is detrimental on nonpathogenic bacteria. Transcriptomics studies reveal that the GATA transcription factor ELT-2 mediates the aberrant immune response upon inhibition of the UFD-1-NPL-4 complex. Collectively, our findings show that inhibition of the UFD-1-NPL-4 complex triggers an aberrant immune response that is detrimental to immunocompetent worms under infection conditions but can be advantageous for immunocompromised worms.

## Introduction

Maintenance of a healthy proteome involves the degradation of misfolded proteins (*Vembar and Brodsky, 2008*; *Vilchez et al., 2014*). Proteasomes are a major site for the degradation of misfolded proteins (*Clague and Urbé, 2010*; *Ding and Yin, 2008*). Proteins are tagged for proteasomal degradation by ubiquitin through a series of enzymatic reactions (*Dikic, 2017*). In addition to the proteasomal degradation of proteins, ubiquitination controls a vast array of cellular signals and functions (*Husnjak and Dikic, 2012*; *Mukhopadhyay and Riezman, 2007*). The ubiquitination pathways also have important roles in innate immunity regulation (*Garcia-Sanchez et al., 2021*; *Li et al., 2016*). The host ubiquitination pathways are involved in the targeting of bacterial proteins, lipopolysaccharides,

and bacteria-containing vacuoles, resulting in selective macroautophagy or xenophagy of bacterial pathogens (*Chai et al., 2019*; *Haldar et al., 2015*; *Otten et al., 2021*; *Tripathi-Giesgen et al., 2021*). Because of the important roles of ubiquitination in host defenses, bacterial pathogens have evolved multiple strategies to target the host ubiquitination pathways (*Bomberger et al., 2011*; *Herhaus and Dikic, 2018*; *Ribet and Cossart, 2018*). Therefore, it is possible that inhibition of the ubiquitination pathways would activate immune responses via compensatory mechanisms.

The proteasomal degradation requires ubiquitin-tagged proteins to be unfolded and extracted from macromolecular complexes (*van den Boom and Meyer, 2018*). CDC-48 (VCP/p97 in vertebrates) is a highly conserved AAA+ATPase that uses its protein unfoldase activity to extract a variety of ubiquitinated polypeptides from membranes or macromolecular complexes (*Bodnar and Rapoport, 2017*; *Meyer et al., 2012*). CDC-48 uses different cofactor proteins to recognize its client proteins (*Meyer et al., 2012*). The UFD-1 (ubiquitin fusion degradation 1)-NPL-4 (nuclear protein localization homolog 4) heterodimer is a CDC-48 cofactor that is involved in the extraction of misfolded and ubiquitinated proteins from several plasma membrane locations, including the endoplasmic reticulum (ER) (*Meyer et al., 2000*; *Wolf and Stolz, 2012*; *Ye et al., 2001*). Therefore, the UFD-1-NPL-4 complex is critical for the ER-associated degradation (ERAD) of misfolded proteins (*Wu and Rapoport, 2018*). In addition to ERAD, the ER has evolved other strategies to deal with misfolded proteins (*Singh, 2023*), including a series of unfolded protein response (UPR) pathways that enhance the folding capacity of the ER by synthesis of molecular chaperones and reduction of protein translation (*Hetz et al., 2015*). The ER-UPR pathways are required for an optimum immune response (*Engel and Barton, 2010*; *Richardson et al., 2010*; *Sun et al., 2012*). However, the role of the UFD-1-NPL-4 complex, which is required for ERAD, in regulating immune responses remains poorly explored. Because ubiquitin-dependent pathways have important roles in immunity, it will be interesting to explore the role of the UFD-1-NPL-4 complex in regulating immune responses.

In this study, we showed that the inhibition of the UFD-1-NPL-4 complex results in the activation of an aberrant immune response in *Caenorhabditis elegans*. The wild-type worms had a significant reduction in survival on the pathogenic bacterium *Pseudomonas aeruginosa* despite having diminished colonization of the gut with the bacterium. Inhibition of the UFD-1-NPL-4 complex also led to diminished bacterial colonization in mutants of ER-UPR and immunity pathways. The diminished bacterial colonization improved the survival of severely immunocompromised mutants on pathogenic bacteria. However, on nonpathogenic bacteria, on which the severely immunocompromised mutants exhibit a normal lifespan, inhibition of the UFD-1-NPL-4 complex resulted in a significant reduction in lifespan. Transcriptomic studies revealed that inhibition of the UFD-1-NPL-4 complex resulted in the activation of the intracellular pathogen response (IPR). Analysis for transcription factor enrichment for upregulated genes identified that the GATA transcription factor ELT-2 mediated the aberrant immune response upon inhibition of the UFD-1-NPL-4 complex. Thus, our studies demonstrated that inhibition of the UFD-1-NPL-4 complex triggers an aberrant immune response that is detrimental to immunocompetent worms under infection conditions but can be advantageous for immunocompromised worms.

## Results

### Inhibition of the UFD-1-NPL-4 complex reduces survival of *C. elegans* on *P. aeruginosa*

To explore the role of the UFD-1-NPL-4 complex in the innate immune response of *C. elegans*, we knocked down *ufd-1* and *npl-4* by RNA interference (RNAi) and studied the survival of the worms on the pathogenic bacterium *P. aeruginosa* PA14. Knockdown of *ufd-1* and *npl-4* resulted in a significant reduction in the survival of N2 wild-type worms on *P. aeruginosa* compared to worms grown on control RNAi (*Figure 1A*). The animals also had a reduced lifespan on *Escherichia coli* HT115 upon knockdown of *ufd-1* and *npl-4* (*Figure 1B*). The relationship between *C. elegans* innate immunity and longevity pathways is complex. Mutants that have reduced lifespan but improved immunity have been identified (*Amrit et al., 2019*; *Otarigho and Aballay, 2021*; *Ren and Ambros, 2015*). Moreover, immunocompromised animals such as mutants of the MAP kinase pathway mediated by NSY-1/SEK-1/PMK-1 have a normal lifespan despite having significantly reduced survival on pathogenic bacteria (*Kim et al., 2002*; *Liberati et al., 2004*). Therefore, we explored the mechanisms that

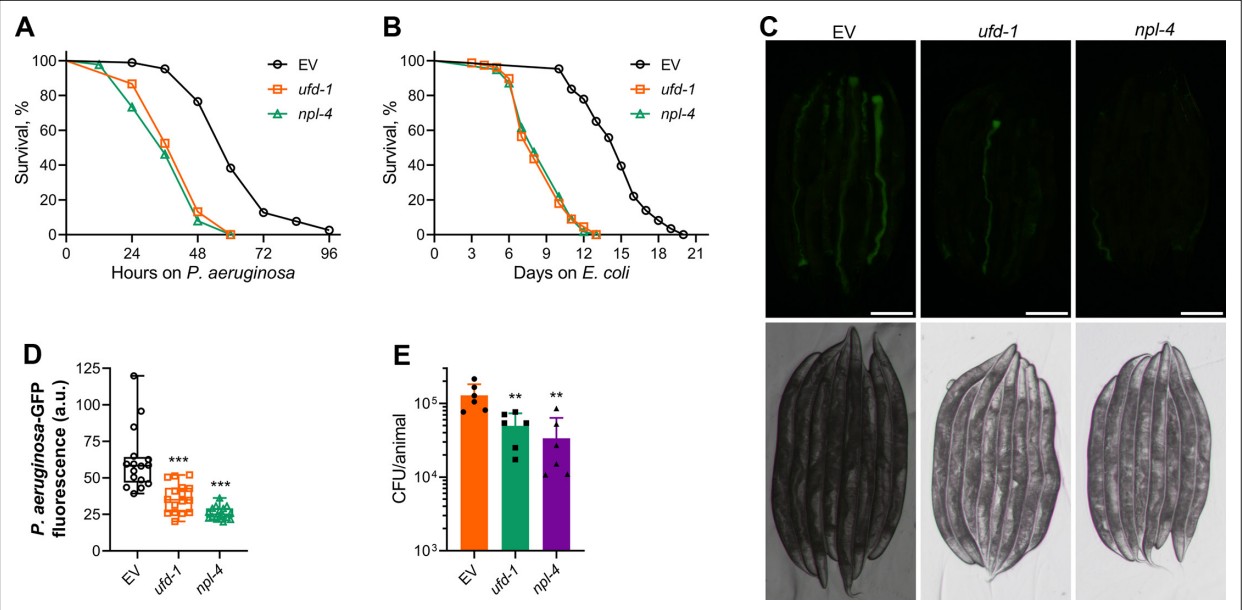

**Figure 1.** Inhibition of the UFD-1-NPL-4 complex reduces survival of *C. elegans* on *P. aeruginosa*. (**A**) Representative survival plots of N2 animals on *P. aeruginosa* PA14 at 25°C after treatment with the empty vector (EV) control, *ufd-1*, and *npl-4* RNA interference (RNAi). p<0.001 for *ufd-1* and *npl-4* RNAi compared to EV control. (**B**) Representative survival plots of N2 animals grown on bacteria for RNAi against *ufd-1* and *npl-4*, along with the EV control at 20°C. Day 0 represents young adults. p<0.001 for *ufd-1* and *npl-4* RNAi compared to EV control. (**C**) Representative fluorescence (top) and the corresponding bright-field (bottom) images of N2 animals incubated on *P. aeruginosa*-GFP for 24 hr at 25°C after growth on the EV control, *ufd-1*, and *npl-4* RNAi bacteria. Scale bar = 200 μm. (**D**) Quantification of GFP levels of N2 animals incubated on *P. aeruginosa*-GFP for 24 hr at 25°C after growth on the EV control, *ufd-1*, and *npl-4* RNAi bacteria. ***p<0.001 via the t-test (n = 16 worms each). (**E**) Colony-forming units (CFUs) per animal of N2 worms incubated on *P. aeruginosa*-GFP for 24 hr at 25°C after growth on the EV control, *ufd-1*, and *npl-4* RNAi bacteria. **p<0.01 via the t-test (n=6 biological replicates).

The online version of this article includes the following source data and figure supplement(s) for figure 1:

**Source data 1.** Inhibition of the UFD-1-NPL-4 complex reduces survival of *C. elegans* on *P. aeruginosa*.

**Figure supplement 1.** RNA interference (RNAi) against *ufd-1* and *npl-4* results in the specific knockdown of their corresponding mRNAs.

**Figure supplement 1—source data 1.** RNA interference (RNAi) against *ufd-1* and *npl-4* results in the specific knockdown of their corresponding mRNAs.

**Figure supplement 2.** Effects on pharyngeal pumping and defecation are unlikely to be the reason for reduced pathogen colonization upon inhibition of the UFD-1-NPL-4 complex.

**Figure supplement 2—source data 1.** Effects on pharyngeal pumping and defecation are unlikely to be the reason for reduced pathogen colonization upon inhibition of the UFD-1-NPL-4 complex.

led to the reduced survival of *ufd-1* and *npl-4* knockdown animals on *P. aeruginosa*. Colonization of the gut with *P. aeruginosa* is a major determinant of infection and survival of *C. elegans* under slow-killing conditions (***Das et al., 2023***; ***Tan et al., 1999***). Because knockdown of *ufd-1* and *npl-4* reduced survival on *P. aeruginosa*, we asked whether the animals had enhanced colonization of the gut with *P. aeruginosa* after *ufd-1* and *npl-4* RNAi. Surprisingly, we observed that *ufd-1* and *npl-4* knockdown resulted in reduced colonization of the gut with *P. aeruginosa* (***Figure 1C and D***). To validate further, we performed the colony-forming unit (CFU) assay upon knockdown of *ufd-1* and *npl-4*. There was a significant decline in the CFU per worm in *ufd-1* and *npl-4* knockdown worms, indicating reduced numbers of live bacteria in these worms (***Figure 1E***).

To confirm the specificity of the RNAi knockdowns and rule out potential off-target effects, we examined transcript levels of *ufd-1* and *npl-4* following RNAi treatment. RNAi against *ufd-1* significantly reduced *ufd-1* mRNA levels without reducing *npl-4* expression, while *npl-4* RNAi specifically downregulated *npl-4* transcripts with no impact on *ufd-1* mRNA levels (***Figure 1—figure supplement 1A and B***). Additionally, alignment of *ufd-1* and *npl-4* mRNA sequences against the *C. elegans* transcriptome revealed no significant similarity to other genes, supporting the specificity of the RNAi constructs. Moreover, the *ufd-1* and *npl-4* RNA sequences do not share significant sequence similarity.

Therefore, the highly similar phenotypes observed in *ufd-1* and *npl-4* knockdown animals, including shortened lifespan, reduced survival on *P. aeruginosa*, and decreased intestinal colonization with *P. aeruginosa*, strongly suggest that these outcomes result from the disruption of the functional UFD-1-NPL-4 complex.

Reduced colonization of the gut by *P. aeruginosa* could be because of reduced uptake of the bacterium in *ufd-1* and *npl-4* knockdown animals. The rate of pharyngeal pumping is an indicator of the uptake of bacterial food. We studied whether inhibition of *ufd-1* and *npl-4* affected the pharyngeal pumping rate. The knockdown of *ufd-1* and *npl-4* did not affect the pharyngeal pumping rate (*Figure 1—figure supplement 2A*), indicating that the reduced colonization was unlikely due to the reduced uptake of bacteria. Next, we tested whether exposure to *P. aeruginosa* affected the pharyngeal pumping in *ufd-1* and *npl-4* knockdown animals. Exposure to *P. aeruginosa* for 12 hr resulted in a minimal reduction in the pharyngeal pumping rate in *ufd-1* and *npl-4* knockdown animals compared to control animals (*Figure 1—figure supplement 2B*). These results suggested that the reduced colonization of *ufd-1* and *npl-4* knockdown animals by *P. aeruginosa* was unlikely because of the reduced uptake of bacteria.

The clearance of intestinal contents through the defecation motor program (DMP) is known to influence gut colonization by *P. aeruginosa* in *C. elegans* (*Das et al., 2023*). It is therefore conceivable that knockdown of the UFD-1-NPL-4 complex might increase defecation frequency, thereby promoting the physical expulsion of bacteria and resulting in reduced gut colonization. To test this possibility, we measured DMP rates in animals subjected to *ufd-1* and *npl-4* RNAi. Contrary to this hypothesis, both *ufd-1* and *npl-4* knockdown animals exhibited a significant reduction in defecation frequency compared to control RNAi-treated animals (*Figure 1—figure supplement 2C*). This reduction in DMP rate persisted even after 12 hr of exposure to *P. aeruginosa* (*Figure 1—figure supplement 2D*). Thus, the change in the DMP rate in *ufd-1* and *npl-4* knockdown animals is unlikely to be the reason for the reduced gut colonization by *P. aeruginosa*.

Taken together, these results suggested that the inhibition of the UFD-1-NPL-4 complex reduced the survival of *C. elegans* on *P. aeruginosa* despite diminished colonization of the gut with the bacterium. Because *ufd-1* and *npl-4* RNAi led to very similar phenotypes, we further used only *ufd-1* RNAi to decipher the mechanisms of reduced survival and diminished colonization on *P. aeruginosa*.

## Reduced colonization with *P. aeruginosa* upon *ufd-1* knockdown is independent of the ER-UPR pathways

Knockdown of the UFD-1-NPL-4 complex is known to cause ER stress, resulting in the upregulation of the ER-UPR pathways (*Mouysset et al., 2006*; *Sasagawa et al., 2007*). Because ER stress and UPR pathways modulate innate immunity (*Richardson et al., 2010*; *Singh and Aballay, 2017*; *Sun et al., 2012*), we asked whether the ER-UPR pathways were involved in the regulation of survival and colonization of *ufd-1* knockdown animals on *P. aeruginosa*. We studied the survival and colonization of mutants of different ER-UPR pathways, including *xbp-1(tm2482)*, *atf-6(ok551)*, and *pek-1(ok275)* on *P. aeruginosa*. The survival of *xbp-1(tm2482)* animals upon *ufd-1* knockdown was indistinguishable from that of the control animals (*Figure 2A*). However, *ufd-1* knockdown resulted in significantly reduced gut colonization with *P. aeruginosa* in *xbp-1(tm2482)* animals (*Figure 2B and C*).

To further examine the role of XBP-1 in this context, we assessed the effect of *ufd-1* knockdown in animals neuronally overexpressing the constitutively active spliced form of XBP-1 (XBP-1s), which has been previously associated with enhanced longevity (*Taylor and Dillin, 2013*). Knockdown of *ufd-1* resulted in the reduced survival of XBP-1s-overexpressing animals on *P. aeruginosa*, despite a concurrent decrease in bacterial colonization of the gut (*Figure 2—figure supplement 1A–C*). This indicated that the XBP-1 pathway was not required for the reduced *P. aeruginosa* colonization of *ufd-1* knockdown animals.

The *atf-6(ok551)* animals exhibited reduced survival on *P. aeruginosa* upon *ufd-1* RNAi compared to the control RNAi (*Figure 2D*). Similar to N2, *ufd-1* knockdown resulted in reduced colonization of the gut with *P. aeruginosa* in *atf-6(ok551)* animals (*Figure 2E and F*). The *pek-1(ok275)* animals exhibited similar phenotypes of reduced survival and diminished colonization on *P. aeruginosa* upon *ufd-1* knockdown (*Figure 2G–I*). These results indicated that the reduced colonization and survival of *ufd-1* knockdown animals on *P. aeruginosa* were independent of the ER-UPR pathways.

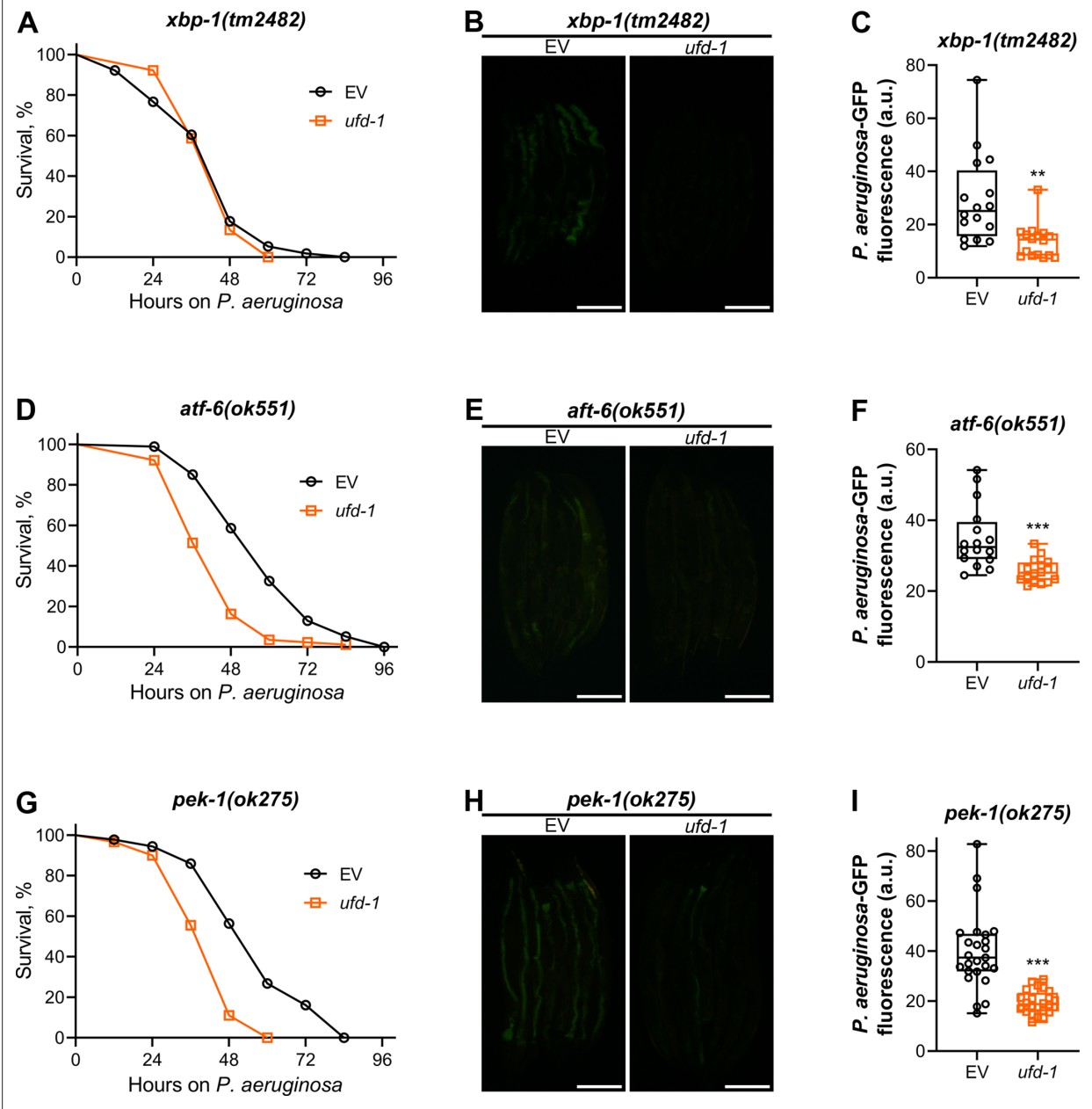

**Figure 2.** Reduced colonization with *P. aeruginosa* upon *ufd-1* knockdown is independent of the ER-UPR pathways. (**A**) Representative survival plots of *xbp-1(tm2482)* animals on *P. aeruginosa* PA14 at 25°C after treatment with the empty vector (EV) control and *ufd-1* RNA interference (RNAi). The difference between the EV and *ufd-1* RNAi survival plots is nonsignificant. (**B**) Representative fluorescence images of *xbp-1(tm2482)* animals incubated on *P. aeruginosa*-GFP for 24 hr at 25°C after growth on the EV control and *ufd-1* RNAi bacteria. Scale bar = 200 µm. (**C**) Quantification of GFP levels of *xbp-1(tm2482)* animals incubated on *P. aeruginosa*-GFP for 24 hr at 25°C after growth on the EV control and *ufd-1* RNAi bacteria. **$p<0.01$ via the t-test (n = 16 worms each). (**D**) Representative survival plots of *atf-6(ok551)* animals on *P. aeruginosa* PA14 at 25°C after treatment with the EV control and *ufd-1* RNAi. $p<0.001$. (**E**) Representative fluorescence images of *atf-6(ok551)* animals incubated on *P. aeruginosa*-GFP for 24 hr at 25°C after growth on the EV control and *ufd-1* RNAi bacteria. Scale bar = 200 µm. (**F**) Quantification of GFP levels of *atf-6(ok551)* animals incubated on *P. aeruginosa*-GFP for 24 hr at 25°C after growth on the EV control and *ufd-1* RNAi bacteria. ***$p<0.001$ via the t-test (n = 16 worms each). (**G**) Representative survival plots of *pek-1(ok275)* animals on *P. aeruginosa* PA14 at 25°C after treatment with the EV control and *ufd-1* RNAi. $p<0.001$. (**H**) Representative fluorescence images of *pek-1(ok275)* animals incubated on *P. aeruginosa*-GFP for 24 hr at 25°C after growth on the EV control and *ufd-1* RNAi bacteria. Scale bar = 200 µm. (**I**) Quantification of GFP levels of *pek-1(ok275)* animals incubated on *P. aeruginosa*-GFP for 24 hr at 25°C after growth on the EV control and *ufd-1* RNAi bacteria. ***$p<0.001$ via the t-test (n = 24–25 worms each).

The online version of this article includes the following source data and figure supplement(s) for figure 2:

**Source data 1.** Reduced colonization with *P. aeruginosa* upon *ufd-1* knockdown is independent of the ER-UPR pathways.

*Figure 2 continued on next page*

*Figure 2 continued*

**Figure supplement 1.** Reduced colonization with *P. aeruginosa* upon *ufd-1* knockdown is independent of the XBP-1 ER-UPR pathway.

**Figure supplement 1—source data 1.** Reduced colonization with *P. aeruginosa* upon *ufd-1* knockdown is independent of the XBP-1 ER-UPR pathway.

## Reduced colonization with *P. aeruginosa* upon *ufd-1* knockdown is independent of the major immunity pathways

Next, we tested whether any of the immunity pathways were involved in regulating the reduced colonization by *ufd-1* knockdown. To this end, we studied the survival and colonization of mutants of different immunity pathways on *P. aeruginosa* upon knockdown of *ufd-1*. The *pmk-1(km25)* animals, which are part of a MAP kinase pathway mediated by NSY-1/SEK-1/PMK-1 (*Kim et al., 2002*), had a similar survival rate on control and *ufd-1* RNAi (*Figure 3A*). However, *ufd-1* knockdown significantly reduced colonization in *pmk-1(km25)* animals (*Figure 3B and C*). This indicated that the *pmk-1* pathway is unlikely to be involved in mediating *ufd-1* knockdown effects, and the similar survival rate of control and *ufd-1* RNAi animals is merely coincidental. Indeed, a mutant of the upstream regulator of the PMK-1 pathway, the Toll/interleukin-1 receptor domain protein (TIR-1) (*Liberati et al., 2004*; *Peterson et al., 2022*), exhibited phenotypes similar to *pmk-1(km25)* animals (*Figure 3—figure supplement 1A–C*).

The mutants of the TGF-β/DBL-1 (*Mallo et al., 2002*) and TFEB/HLH-30 (*Visvikis et al., 2014*) immunity pathways also exhibited phenotypes similar to *pmk-1(km25)* animals. The knockdown of *ufd-1* resulted in significantly reduced colonization of *dbl-1(nk3)* and *hlh-30(tm1978)* animals without affecting their survival on *P. aeruginosa* (*Figure 3D–I*). These results indicated that the reduced colonization with *P. aeruginosa* upon *ufd-1* knockdown was independent of these immunity pathways.

## Inhibition of the UFD-1-NPL-4 complex improves survival of severely immunocompromised *C. elegans* on *P. aeruginosa*

Knockdown of *ufd-1* reduced colonization in wild-type animals, as well as mutants of ER-UPR and innate immunity pathways. Interestingly, despite variable survival rates on control RNAi, all strains had similar survival rates upon knockdown of *ufd-1* (*Figure 4—figure supplement 1*). We reasoned that *ufd-1* knockdown might lead to an aberrant immune response that results in diminished gut colonization by *P. aeruginosa* and reduces the survival of healthy but not immunocompromised animals. If this were the case, knockdown of the UFD-1-NPL-4 complex might improve the survival of severely immunocompromised animals by activating a compensatory immune response. In *C. elegans*, the canonical p38 MAP kinase signaling cascade consists of NSY-1 (ASK1 MAPKKK), SEK-1 (MKK3/MKK6 MAPKK), and PMK-1 (p38 MAPK) (*Kim et al., 2002*). Compared to *pmk-1* knockout, the knockout of *sek-1* leads to more severe effects on survival upon infection with *P. aeruginosa* (*Meng et al., 2021*). Indeed, we observed that most of the *sek-1(km4)* animals on control RNAi died within 24 hr of exposure to *P. aeruginosa* (*Figure 4A*). Importantly, the knockdown of *ufd-1* resulted in a significant improvement in the survival of *sek-1(km4)* animals (*Figure 4A*). To test whether the improved survival of *sek-1(km4)* animals upon the knockdown of *ufd-1* was because of inhibition of the UFD-1-NPL-4 complex, we studied the survival of *sek-1(km4)* animals upon *npl-4* RNAi. The knockdown of *npl-4* also resulted in significantly enhanced survival of *sek-1(km4)* animals (*Figure 4A*).

Knockdown of both *ufd-1* and *npl-4* in *sek-1(km4)* animals resulted in reduced gut colonization by *P. aeruginosa* compared to control RNAi (*Figure 4B and C*). Because *sek-1(km4)* animals do not have a reduced lifespan on *E. coli* diet (*Kim et al., 2002*), we studied how inhibition of the UFD-1-NPL-4 complex affected the lifespan of *sek-1(km4)* animals. We reasoned that if the inhibition of the UFD-1-NPL-4 complex resulted in a heightened aberrant immune response, it should result in a reduced lifespan of *sek-1(km4)* animals on *E. coli* despite improving their survival on *P. aeruginosa*. Indeed, the knockdown of *ufd-1* and *npl-4* significantly reduced the lifespan of *sek-1(km4)* animals on *E. coli* HT115 (*Figure 4D*). These results suggested that inhibition of the UFD-1-NPL-4 complex led to an aberrant immune response, which improves survival of severely immunocompromised animals under infection conditions but reduces survival of such animals under non-infection conditions.

To further establish that inhibition of the UFD-1-NPL-4 complex resulted in the improved survival of severely immunocompromised animals on *P. aeruginosa*, we created a *dbl-1(nk3);pmk-1(km25)* double mutant. The TGF-β/DBL-1 and p38 MAPK/PMK-1 control immunity via parallel pathways (*Singh and*

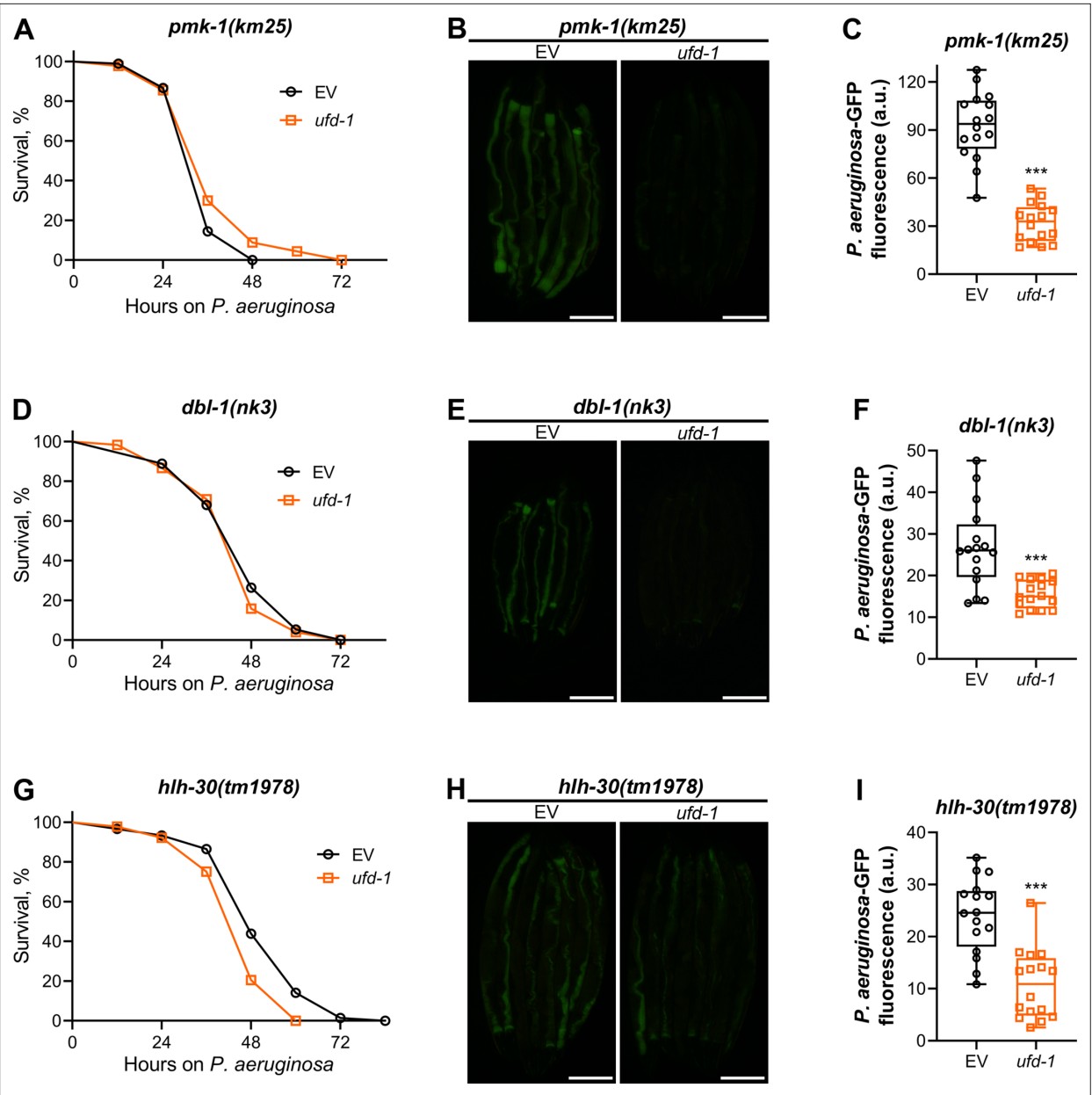

**Figure 3.** Reduced colonization with *P. aeruginosa* upon *ufd-1* knockdown is independent of the major immunity pathways. (**A**) Representative survival plots of *pmk-1(km25)* animals on *P. aeruginosa* PA14 at 25°C after treatment with the empty vector (EV) control and *ufd-1* RNA interference (RNAi). The difference between the EV and *ufd-1* RNAi survival plots is nonsignificant. (**B**) Representative fluorescence images of *pmk-1(km25)* animals incubated on *P. aeruginosa*-GFP for 24 hr at 25°C after growth on the EV control and *ufd-1* RNAi bacteria. Scale bar = 200 μm. (**C**) Quantification of GFP levels of *pmk-1(km25)* animals incubated on *P. aeruginosa*-GFP for 24 hr at 25°C after growth on the EV control and *ufd-1* RNAi bacteria. ***p<0.001 via the t-test (n = 16 worms each). (**D**) Representative survival plots of *dbl-1(nk3)* animals on *P. aeruginosa* PA14 at 25°C after treatment with the EV control and *ufd-1* RNAi. The difference between the EV and *ufd-1* RNAi survival plots is nonsignificant. (**E**) Representative fluorescence images of *dbl-1(nk3)* animals incubated on *P. aeruginosa*-GFP for 24 hr at 25°C after growth on the EV control and *ufd-1* RNAi bacteria. Scale bar = 200 μm. (**F**) Quantification of GFP levels of *dbl-1(nk3)* animals incubated on *P. aeruginosa*-GFP for 24 hr at 25°C after growth on the EV control and *ufd-1* RNAi bacteria. ***p<0.001 via the t-test (n = 16 worms each). (**G**) Representative survival plots of *hlh-30(tm1978)* animals on *P. aeruginosa* PA14 at 25°C after treatment with the EV control and *ufd-1* RNAi. p<0.001. (**H**) Representative fluorescence images of *hlh-30(tm1978)* animals incubated on *P. aeruginosa*-GFP for 24 hr at 25°C after growth on the EV control and *ufd-1* RNAi bacteria. Scale bar = 200 μm. (**I**) Quantification of GFP levels of *hlh-30(tm1978)* animals incubated on *P. aeruginosa*-GFP for 24 hr at 25°C after growth on the EV control and *ufd-1* RNAi bacteria. ***p<0.001 via the t-test (n = 16 worms each).

The online version of this article includes the following source data and figure supplement(s) for figure 3:

**Source data 1.** Reduced colonization with *P. aeruginosa* upon *ufd-1* knockdown is independent of the major immunity pathways.

*Figure 3 continued on next page*

*Figure 3 continued*

**Figure supplement 1.** Reduced gut colonization upon *ufd-1* knockdown is independent of the *tir-1* immunity pathway.

**Figure supplement 1—source data 1.** Reduced gut colonization upon *ufd-1* knockdown is independent of the *tir-1* immunity pathway.

*Aballay, 2020*), and the *dbl-1(nk3);pmk-1(km25)* animals show reduced survival on *P. aeruginosa* compared to individual mutants (*Figure 4E*). The knockdown of *ufd-1* resulted in improved survival (*Figure 4E*) and reduced colonization (*Figure 4F and G*) of *dbl-1(nk3);pmk-1(km25)* animals on *P. aeruginosa*. Taken together, these data showed that inhibition of the UFD-1-NPL-4 complex improved the survival of severely immunocompromised animals on *P. aeruginosa*.

## Knockdown of *ufd-1* results in the upregulation of protease and IPR genes

To understand the molecular basis of the phenotypes observed upon knockdown of *ufd-1*, we used RNA sequencing to focus on transcriptional changes induced by *ufd-1* RNAi. Of the 439 differentially regulated genes, 319 were upregulated, while 120 were downregulated (*Figure 5A* and *Supplementary file 1*). Gene ontology (GO) analysis for biological processes for upregulated genes showed enrichment for innate immune response (*Figure 5B*). As *ufd-1* is required for ERAD, different ER-UPR pathway genes were also enriched in the upregulated genes. In addition, enrichment for proteolysis genes was also observed. GO analysis for cellular components and molecular function for upregulated

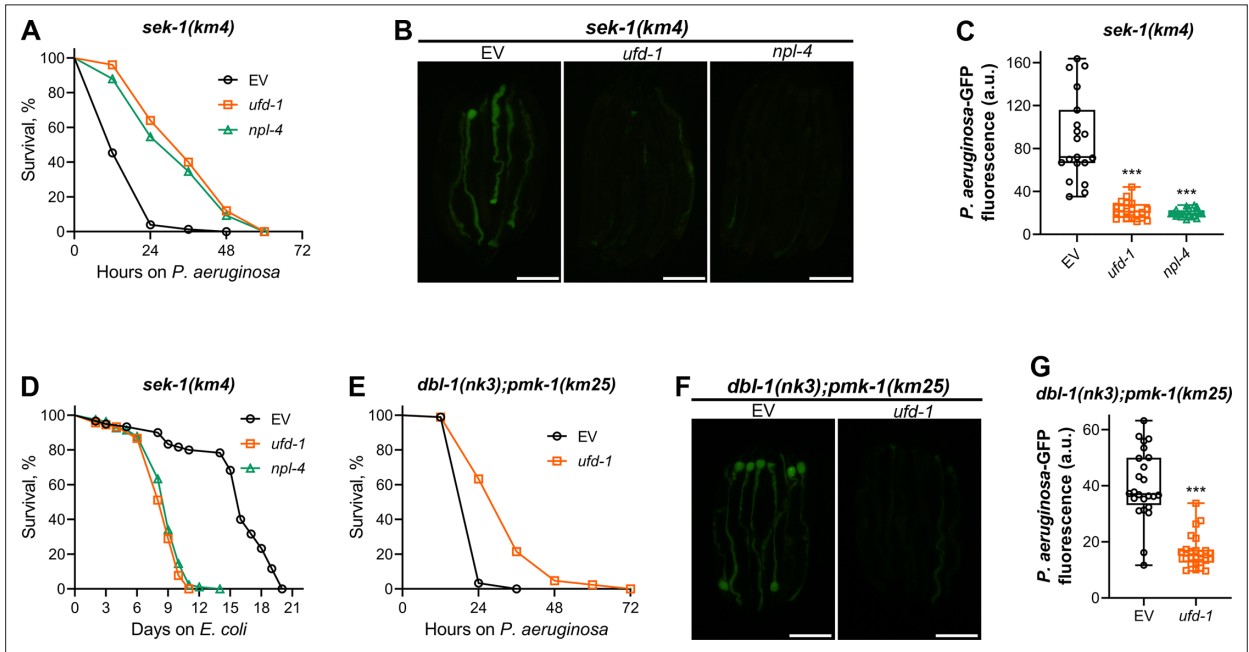

**Figure 4.** Inhibition of the UFD-1-NPL-4 complex improves survival of severely immunocompromised *C. elegans* on *P. aeruginosa*. (**A**) Representative survival plots of *sek-1(km4)* animals on *P. aeruginosa* PA14 at 25°C after treatment with the empty vector (EV) control, *ufd-1*, and *npl-4* RNA interference (RNAi). p<0.001 for *ufd-1* and *npl-4* RNAi compared to EV control. (**B**) Representative fluorescence images of *sek-1(km4)* animals incubated on *P. aeruginosa*-GFP for 12 hr at 25°C after growth on the EV control, *ufd-1*, and *npl-4* RNAi bacteria. Scale bar = 200 µm. (**C**) Quantification of GFP levels of *sek-1(km4)* animals incubated on *P. aeruginosa*-GFP for 12 hr at 25°C after growth on the EV control, *ufd-1*, and *npl-4* RNAi bacteria. ***p<0.001 via the t-test (n = 19–20 worms each). (**D**) Representative survival plots of *sek-1(km4)* animals grown on bacteria for RNAi against *ufd-1* and *npl-4* along with the EV control at 20°C. Day 0 represents young adults. p<0.001 for *ufd-1* and *npl-4* RNAi compared to EV control. (**E**) Representative survival plots of *dbl-1(nk3);pmk-1(km25)* animals on *P. aeruginosa* PA14 at 25°C after treatment with the EV control and *ufd-1* RNAi. p<0.001. (**F**) Representative fluorescence images of *dbl-1(nk3);pmk-1(km25)* animals incubated on *P. aeruginosa*-GFP for 12 hr at 25°C after growth on the EV control and *ufd-1* RNAi bacteria. Scale bar = 200 µm. (**G**) Quantification of GFP levels of *dbl-1(nk3);pmk-1(km25)* animals incubated on *P. aeruginosa*-GFP for 12 hr at 25°C after growth on the EV control and *ufd-1* RNAi bacteria. ***p<0.001 via the t-test (n = 24 worms each).

The online version of this article includes the following source data and figure supplement(s) for figure 4:

**Source data 1.** Inhibition of the UFD-1-NPL-4 complex improves survival of severely immunocompromised *C. elegans* on *P. aeruginosa*.

**Figure supplement 1.** Knockdown of *ufd-1* dictates survival of different worm strains on *P. aeruginosa*.

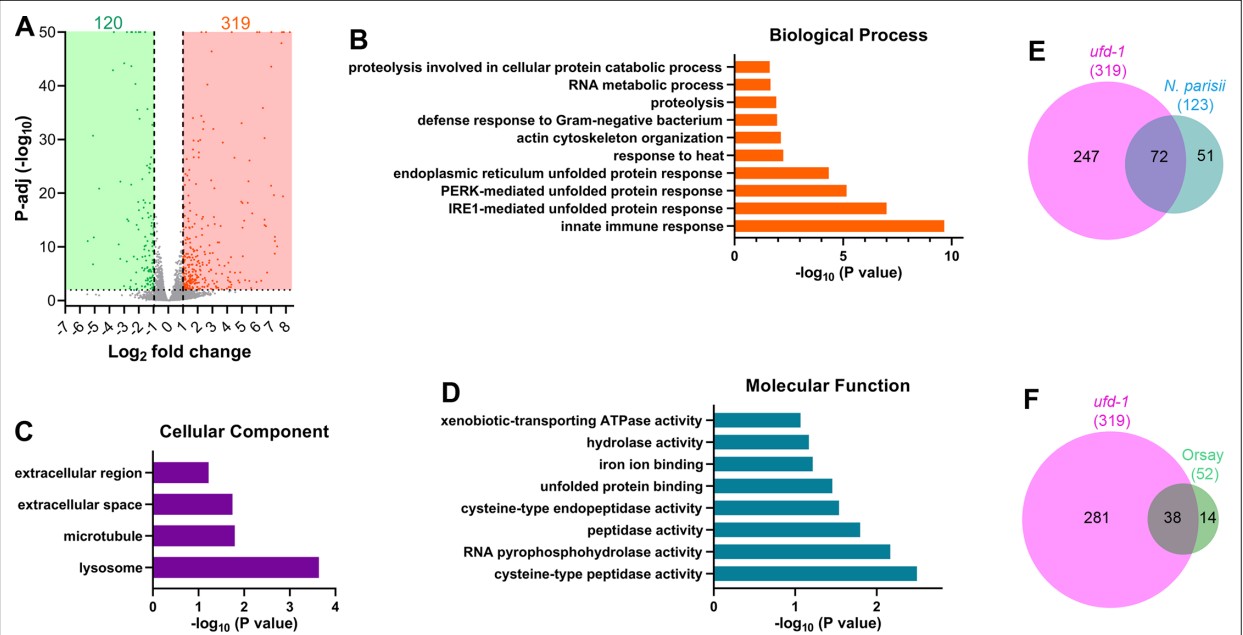

**Figure 5.** Knockdown of *ufd-1* results in the upregulation of protease and intracellular pathogen response genes. (**A**) Volcano plot of upregulated and downregulated genes in *ufd-1* RNA interference (RNAi) versus empty vector (EV) control RNAi N2 animals. Orange and green dots represent significantly upregulated and downregulated genes, respectively, while the gray dots represent the genes that are not differentially regulated. (**B–D**) Gene ontology enrichment analysis for *ufd-1* RNAi upregulated genes for biological processes (**B**), cellular component (**C**), and molecular function (**D**). (**E**) Venn diagram showing the overlap between genes upregulated upon *ufd-1* RNAi and upregulated upon *Nematocida parisii* infection (***Bakowski et al., 2014***). The p-value for the overlap between the data is $8.2 \times 10^{-110}$. (**F**) Venn diagram showing the overlap between genes upregulated upon *ufd-1* RNAi and upregulated upon Orsay virus infection (***Sarkies et al., 2013***). The p-value for the overlap between the data is $2.7 \times 10^{-62}$.

The online version of this article includes the following figure supplement(s) for figure 5:

**Figure supplement 1.** Gene ontology enrichment analysis for *ufd-1* RNA interference (RNAi) downregulated genes.

genes showed enrichment for lysosomes and peptidase activities, respectively (***Figure 5C and D***). These results indicated that the knockdown of *ufd-1* might result in increased proteolysis activities via lysosomes. GO analysis for biological processes for downregulated genes also showed enrichment for innate immune response (***Figure 5—figure supplement 1***). This indicated that *ufd-1* might be required for the expression of some innate immune response genes. GO analysis for cellular components and molecular function for downregulated genes primarily identified functions related to nucleosomes (***Figure 5—figure supplement 1B and C***). Indeed, UFD-1 is known to localize to the nucleus and is required for chromatin stability (***Mouysset et al., 2008***).

Next, we compared the upregulated genes with previously published gene expression data using WormExp (***Yang et al., 2016b***). Interestingly, we observed that the *ufd-1* RNAi upregulated genes had a very high overlap with genes upregulated by intracellular pathogens *N. parisii* (***Bakowski et al., 2014***; ***Figure 5E***) and Orsay virus (***Sarkies et al., 2013***; ***Figure 5F***). Infection with the intracellular pathogens *N. parisii* and Orsay virus results in the activation of an IPR, which includes several protein-containing *ALS*2CR12 signature (*pals*) genes, as well as genes involved in proteolysis (***Bakowski et al., 2014***; ***Sarkies et al., 2013***). Indeed, *ufd-1* RNAi resulted in the upregulation of several *pals* and proteolysis genes (***Supplementary file 1***). These results indicated that the knockdown of *ufd-1* might mimic an intracellular pathogen infection and result in the activation of the IPR.

## GATA transcription factor ELT-2 mediates the *ufd-1* knockdown phenotypes

To identify the genes downstream of *ufd-1* knockdown that were responsible for reduced colonization, we knocked down individual genes that were upregulated upon *ufd-1* RNAi and studied colonization of the gut with *P. aeruginosa*. We hypothesized that the reduced colonization might be because of increased expression of proteolysis genes. Therefore, we knocked down the proteolysis genes that

were upregulated by *ufd-1* RNAi. Knockdown of the protease genes *cpr-1*, *asp-12*, and Y71H2AR.25 led to a significant increase in colonization in *ufd-1* knockdown animals (***Figure 6—figure supplement 1A***), suggesting that increased expression of protease genes might be responsible for reduced colonization. Because *ufd-1* knockdown also resulted in the upregulation of several *pals* genes, which are part of the IPR, we next targeted the *pals* genes upregulated by *ufd-1* RNAi. Knockdown of the *pals* genes, *pals-9*, *pals-14*, *pals-16*, *pals-17*, and *pals-29*, resulted in a significant increase in colonization in *ufd-1* knockdown animals (***Figure 6—figure supplement 1B***). These results indicated that some of the proteases and IPR genes are involved in the regulation of colonization in *ufd-1* knockdown animals. It is also likely that these genes function redundantly, and multiple genes contribute to the observed phenotype.

To identify the transcription factors that regulate the diminished colonization and reduced survival phenotype of *ufd-1* knockdown, we carried out transcription factor enrichment analysis for upregulated genes using WormExp. The *ufd-1* RNAi upregulated genes substantially overlapped with the genes regulated by the GATA transcription factor ELT-2 (***Dineen et al., 2018***; ***Mann et al., 2016***; ***Figure 6A and B***). Indeed, ELT-2 is known to regulate the expression of protease and *pals* genes (***Mann et al., 2016***). To assess whether ELT-2 contributes to the phenotypes associated with *ufd-1* knockdown, we conducted double RNAi experiments in N2 animals targeting both *ufd-1* and *elt-2* and examined *P. aeruginosa* colonization. The double knockdown of *ufd-1* and *elt-2* did not compromise RNAi efficiency, as evidenced by significantly reduced *ufd-1* mRNA levels and diminished ELT-2::GFP signal (***Figure 6—figure supplement 2A and B***). Importantly, knockdown of *elt-2* resulted in a significant increase in *P. aeruginosa* colonization in *ufd-1* knockdown animals (***Figure 6C and D***). Similar to N2 worms, the knockdown of *elt-2* resulted in a significant increase in *P. aeruginosa* colonization in *ufd-1* knockdown *sek-1(km4)* animals (***Figure 6E and F***). Because *ufd-1* knockdown in *sek-1(km4)* worms improves their survival on *P. aeruginosa* (***Figure 4A***), we also studied whether *elt-2* was responsible for the increased survival of *sek-1(km4)* worms upon *ufd-1* knockdown. Indeed, the knockdown of *elt-2* abolished the beneficial effects of *ufd-1* knockdown on the survival of *sek-1(km4)* worms on *P. aeruginosa* (***Figure 6G***).

To determine whether ELT-2 activation alone is sufficient to recapitulate the phenotypes observed upon UFD-1-NPL-4 complex inhibition, we analyzed animals overexpressing ELT-2. Similar to *ufd-1* knockdown, ELT-2 overexpression led to a significant reduction in the colonization of the gut by *P. aeruginosa* (***Figure 6—figure supplement 3A and B***). However, overexpression of ELT-2 did not alter the survival of worms on *P. aeruginosa* (***Figure 6—figure supplement 3C***). Taken together, these findings suggest that the phenotypes triggered by disruption of the UFD-1-NPL-4 complex are partially mediated by ELT-2. However, additional pathways, yet to be identified, likely cooperate with ELT-2 to regulate both pathogen resistance and host survival.

## Discussion

In this study, we show that inhibition of the UFD-1-NPL-4 complex leads to an aberrant immune response in *C. elegans*. Suppression of this complex leads to reduced gut colonization with *P. aeruginosa* in wild-type animals, as well as mutants of different ER-UPR and immunity pathways. Despite the reduction in pathogen load, the wild-type animals exhibit reduced survival, indicative of a detrimental immune response. However, immunocompromised mutants, which have significantly reduced survival on pathogenic bacteria, exhibit reduced colonization and improved survival on inhibition of the UFD-1-NPL-4 complex. This indicates that the immune response activated by the inhibition of the UFD-1-NPL-4 complex compensates for the dampened immune response of the immunocompromised mutants. Despite beneficial effects on the survival of pathogenic bacteria, inhibition of the UFD-1-NPL-4 complex leads to adverse effects on the lifespan of immunocompromised mutants. This is because the immunocompromised mutants have a normal lifespan on nonpathogenic bacteria, and the aberrant immune response becomes detrimental on such bacterial diets.

Previous studies have shown that hyperactivation of immune pathways can negatively affect organismal development. For example, sustained activation of the p38 MAPK pathway impairs development in *C. elegans* (***Cheesman et al., 2016***; ***Kim et al., 2016***), and excessive activation of the IPR also leads to developmental defects (***Lažetić et al., 2023***). Similar to our current study, recent work has demonstrated that heightened immune responses can reduce gut pathogen load while paradoxically decreasing host survival during infection (***Ghosh and Singh, 2024***; ***Peterson et al., 2022***). However,

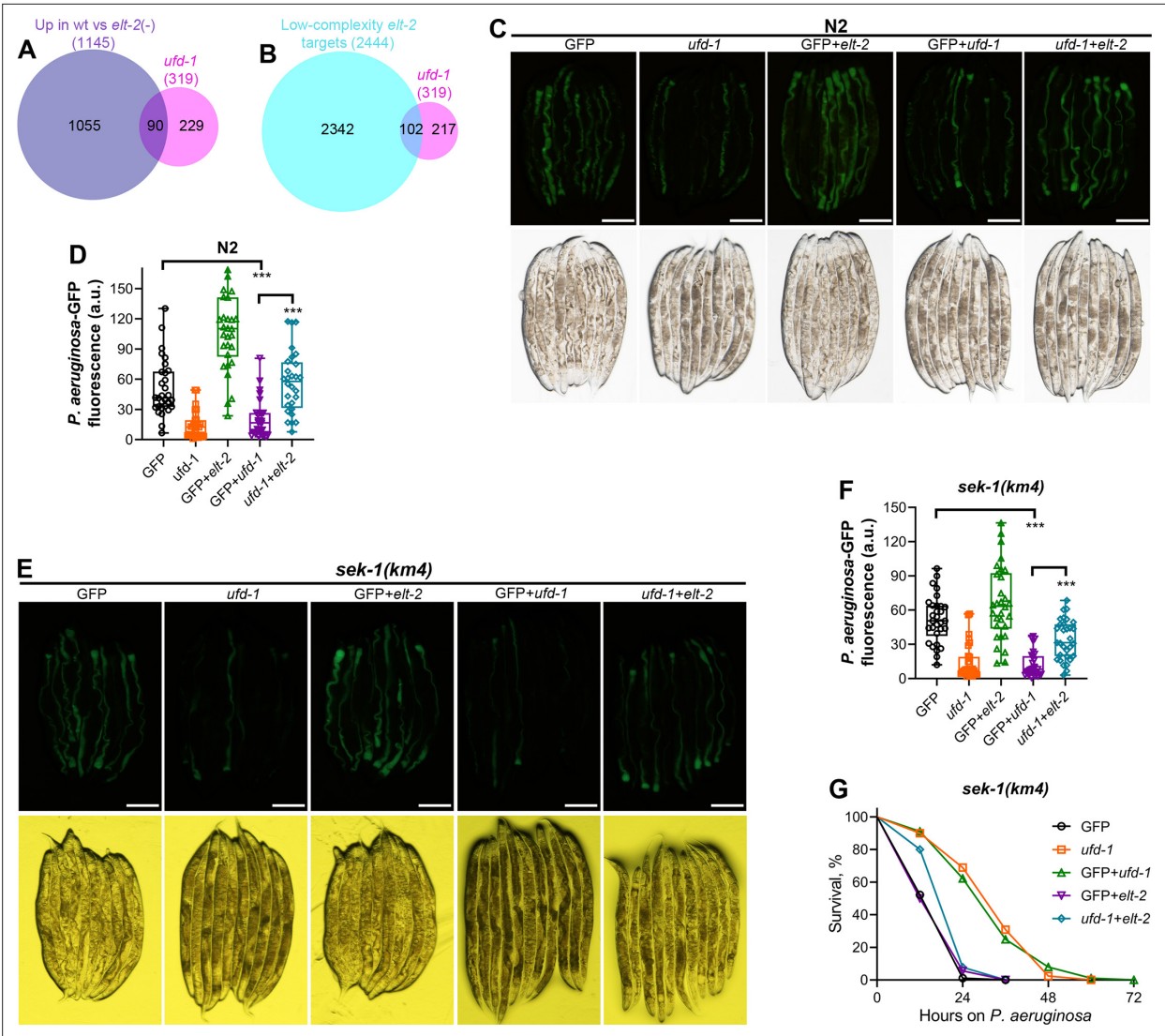

**Figure 6.** GATA transcription factor ELT-2 mediates the *ufd-1* knockdown phenotypes. (**A**) Venn diagram showing the overlap between genes upregulated upon *ufd-1* RNA interference (RNAi) and upregulated in wt versus *elt-2*(-) larvae (***Dineen et al., 2018***). The p-value for the overlap between the data is $9.5 \times 10^{-52}$. (**B**) Venn diagram showing the overlap between genes upregulated upon *ufd-1* RNAi and the low-complexity ELT-2 target genes (***Mann et al., 2016***). The p-value for the overlap between the data is $1.5 \times 10^{-34}$. (**C**) Representative fluorescence (top) and the corresponding bright-field (bottom) images of N2 animals incubated on *P. aeruginosa*-GFP for 24 hr at 25°C after growth on the *gfp* RNAi (GFP) control, *ufd-1*, GFP+*elt-2*, GFP+*ufd-1*, and *ufd-1*+*elt-2* RNAi bacteria (see Materials and methods for the details). Scale bar = 200 μm. (**D**) Quantification of GFP levels of N2 animals incubated on *P. aeruginosa*-GFP for 24 hr at 25°C after growth on the *gfp* RNAi control, *ufd-1*, GFP+*elt-2*, GFP+*ufd-1*, and *ufd-1*+*elt-2* RNAi bacteria. ***p<0.001 via the t-test (n = 28–30 worms each). (**E**) Representative fluorescence (top) and the corresponding bright-field (bottom) images of *sek-1(km4)* animals incubated on *P. aeruginosa*-GFP for 12 hr at 25°C after growth on the *gfp* RNAi control, *ufd-1*, GFP+*elt-2*, GFP+*ufd-1*, and *ufd-1*+*elt-2* RNAi bacteria. Scale bar = 200 μm. (**F**) Quantification of GFP levels of *sek-1(km4)* animals incubated on *P. aeruginosa*-GFP for 12 hr at 25°C after growth on the *gfp* RNAi control, *ufd-1*, GFP+*elt-2*, GFP+*ufd-1*, and *ufd-1*+*elt-2* RNAi bacteria. ***p<0.001 via the t-test (n = 30–31 worms each). (**G**) Representative survival plots of *sek-1(km4)* animals on *P. aeruginosa* PA14 at 25°C after treatment with the *gfp* RNAi control, *ufd-1*, GFP+*elt-2*, GFP+*ufd-1*, and *ufd-1*+*elt-2* RNAi bacteria. p<0.001 for *ufd-1*+*elt-2* RNAi compared to GFP+*ufd-1* RNAi.

The online version of this article includes the following source data and figure supplement(s) for figure 6:

**Source data 1.** GATA transcription factor ELT-2 mediates the *ufd-1* knockdown phenotypes.

**Figure supplement 1.** Role of protease or intracellular pathogen response genes in reduced colonization of *ufd-1* RNA interference (RNAi) animals.

**Figure supplement 1—source data 1.** Role of protease or intracellular pathogen response genes in reduced colonization of *ufd-1* RNA interference (RNAi) animals.

**Figure supplement 2.** Double RNA interference (RNAi) does not impact *ufd-1* or*elt-2* mRNA knockdown.

*Figure 6 continued on next page*

*Figure 6 continued*

**Figure supplement 2—source data 1.** Double RNA interference (RNAi) does not impact *ufd-1* or *elt-2* mRNA knockdown.

**Figure supplement 3.** ELT-2 overexpression partly recapitulates the *ufd-1* knockdown phenotypes.

**Figure supplement 3—source data 1.** ELT-2 overexpression partly recapitulates the *ufd-1* knockdown phenotypes.

our study uniquely shows that while such heightened immune responses are detrimental to immuno-competent animals, they can be beneficial in the context of immunodeficiency.

We find that the knockdown of *ufd-1* led to the upregulation of several genes that are part of the IPR activated by intracellular pathogens *N. parisii* and Orsay virus (*Bakowski et al., 2014*; *Sarkies et al., 2013*). The ubiquitination components have been shown to be required for targeting the intracellular pathogen *N. parisii* (*Bakowski et al., 2014*). As a counterattack, the pathogen probably targets the ubiquitin-proteasome system of the host (*Bakowski et al., 2014*). Therefore, the intracellular pathogens *N. parisii* and Orsay virus might activate the IPR by inhibiting the ubiquitin-proteasome system (*Bakowski et al., 2014*; *Reddy et al., 2019*). We show that inhibition of the UFD-1-NPL-4 complex activates the IPR. The activation of the IPR by the inhibition of the UFD-1-NPL-4 complex could be a consequence of the direct inhibition of this complex itself or an indirect perturbation of the proteasomal degradation of proteins. In future studies, it will be intriguing to decipher whether intracellular pathogens target the UFD-1-NPL-4 complex.

We demonstrate that the GATA transcription factor ELT-2 mediated the aberrant response downstream of the inhibition of the UFD-1-NPL-4 complex. ELT-2 is known to be required for defense, as well as recovery responses against a variety of pathogens (*Head and Aballay, 2014*; *Shapira et al., 2006*; *Yang et al., 2016a*). Previous studies have reported interactions of ELT-2 with the proteasome system. The non-proteolytic activity of the 19S proteasome subunit RPT-6 was shown to regulate the ELT-2-mediated immune response (*Olaitan and Aballay, 2018*). Another study showed that the bacterial pathogen *Burkholderia pseudomallei* leads to the downregulation of ELT-2 target genes (*Lee et al., 2013*). It was demonstrated that the downregulation of ELT-2 targets was associated with the degradation of ELT-2 protein by the host ubiquitin-proteasome system. Therefore, multiple mechanisms could regulate the activity of ELT-2 via the ubiquitin-proteasome system. In future studies, it will be interesting to study how inhibition of the UFD-1-NPL-4 complex modulates the activities of the GATA transcription factor ELT-2.

## Materials and methods

### Key resources table

| Reagent type (species) or resource | Designation | Source or reference | Identifiers | Additional information |
|---|---|---|---|---|
| Strain, strain background (*Escherichia coli*) | OP50 | Caenorhabditis Genetics Center (CGC) | OP50 | |
| Strain, strain background (*E. coli*) | HT115(DE3) | Source BioScience | HT115(DE3) | |
| Strain, strain background (*Pseudomonas aeruginosa*) | PA14 | Frederick M Ausubel laboratory | PA14 | |
| Strain, strain background (*P. aeruginosa*) | PA14-GFP | Frederick M Ausubel laboratory | PA14-GFP | |
| Strain, strain background (*Caenorhabditis elegans*) | N2 Bristol | CGC | N2 | |

*Continued on next page*

*Continued*

| Reagent type (species) or resource | Designation | Source or reference | Identifiers | Additional information |
|---|---|---|---|---|
| Strain, strain background (*C. elegans*) | *sek-1(km4)* | CGC | KU4 | |
| Strain, strain background (*C. elegans*) | *pmk-1(km25)* | CGC | KU25 | |
| Strain, strain background (*C. elegans*) | *dbl-1(nk3)* | CGC | NU3 | |
| Strain, strain background (*C. elegans*) | *hlh-30(tm1978)* | CGC | JIN1375 | |
| Strain, strain background (*C. elegans*) | *xbp-1(tm2482)* | NBRP, Japan | *xbp-1(tm2482)* | |
| Strain, strain background (*C. elegans*) | *pek-1(ok275)* | CGC | RB545 | |
| Strain, strain background (*C. elegans*) | *atf-6(ok551)* | CGC | RB772 | |
| Strain, strain background (*C. elegans*) | *tir-1(qd4)* | CGC | RB1085 | |
| Strain, strain background (*C. elegans*) | *uthIs270 [rab-3p::xbp-1s (constitutively active)+myo-2p::tdTomato]* | CGC | AGD927 | |
| Strain, strain background (*C. elegans*) | *glo-4(ok623); gaIs290 [elt-2::TY1::EGFP::3xFLAG(92C12)+unc-119(+)]* | CGC | SD1949 | |
| Strain, strain background (*C. elegans*) | *dbl-1(nk3);pmk-1(km25)* | This study | *dbl-1(nk3);pmk-1(km25)* | Materials and methods section |
| Strain, strain background (*C. elegans*) | *jsnEx3 [elt-2p::elt-2+myo-2p::mCherry]* | This study | *elt-2_OE* | Materials and methods section |
| Sequence-based reagent | Pan-act_qPCR_F | This study | qPCR primers | TCGGTATGGGACAGAAGGAC |
| Sequence-based reagent | Pan-act_qPCR_R | This study | qPCR primers | CATCCCAGTTGGTGACGATA |
| Sequence-based reagent | npl-4_qPCR_F | This study | qPCR primers | AATGGAGGAAGCGGCAATGA |
| Sequence-based reagent | npl-4_qPCR_R | This study | qPCR primers | TCCACAGTTCCACACAGCTC |
| Sequence-based reagent | ufd-1_qPCR_F | This study | qPCR primers | GGTCGTGTTTCATTCCTTCG |
| Sequence-based reagent | ufd-1_qPCR_R | This study | qPCR primers | TTGCCTCCACGGAAGACATT |
| Sequence-based reagent | npl-4_RNAi_F | This study | Cloning primers | GCT<u>CCCGGG</u>ATGGTACTTGAAGTCCCTCA |

*Continued on next page*

*Continued*

| Reagent type (species) or resource | Designation | Source or reference | Identifiers | Additional information |
|---|---|---|---|---|
| Sequence-based reagent | npl-4_RNAi_R | This study | Cloning primers | AGGTCTAGAATCGGCAGCTGGCAATCCAC |
| Sequence-based reagent | elt-2_OE_F | This study | Cloning primers | CGTCTGCAG CTGATTGTTTCAGAACACCC |
| Sequence-based reagent | elt-2_OE_R | This study | Cloning primers | CGACCCGGG AAGTAGGGTACACATGTTTG |
| Software, algorithm | GraphPad Prism 8 | GraphPad Software | RRID:SCR_002798 | https://www.graphpad.com/scientificsoftware/prism/ |
| Software, algorithm | Photoshop CS5 | Adobe | RRID:SCR_014199 | https://www.adobe.com/products/photoshop.html |
| Software, algorithm | ImageJ | NIH | RRID:SCR_003070 | https://imagej.nih.gov/ij/ |

## Bacterial strains

The following bacterial strains were used in the current study: *E. coli* OP50, *E. coli* HT115(DE3), *P. aeruginosa* PA14, and *P. aeruginosa* PA14 expressing green fluorescent protein (*P. aeruginosa* PA14-GFP). The cultures of *E. coli* OP50, *E. coli* HT115(DE3), and *P. aeruginosa* PA14 were grown in Luria-Bertani (LB) broth at 37°C. The *P. aeruginosa* PA14-GFP cultures were grown in LB broth with 50 µg/mL kanamycin at 37°C.

## *C. elegans* strains and growth conditions

*C. elegans* hermaphrodites were maintained at 20°C on nematode growth medium (NGM) plates seeded with *E. coli* OP50 as the food source unless otherwise indicated. Bristol N2 was used as the wild-type control unless otherwise indicated. The following strains were used in the study: KU4 *sek-1(km4)*, KU25 *pmk-1(km25)*, NU3 *dbl-1(nk3)*, JIN1375 *hlh-30(tm1978)*, *xbp-1(tm2482)*, RB545 *pek-1(ok275)*, RB772 *atf-6(ok551)*, and RB1085 *tir-1(qd4)*, AGD927 *uthIs270 [rab-3p::xbp-1s (constitutively active)+myo-2p::tdTomato]*, and SD1949 *glo-4(ok623);gaIs290 [elt-2::TY1::EGFP::3xFLAG(92C12)+unc-119(+)]*. Some of the strains were obtained from the Caenorhabditis Genetics Center (University of Minnesota, Minneapolis, MN, USA). The *dbl-1(nk3);pmk-1(km25)* strain was obtained by a standard genetic cross.

## Construction of the *npl-4* RNAi clone

The 1581-base-pair full-length cDNA of *npl-4.1* was amplified using the forward primer 5'- GCTCCCGGGATGGTACTTGAAGTCCCTCA -3' and the reverse primer 5'- AGGTCTAGAATCGGCAGCTGGCAATCCAC -3'. Because the nucleotide sequence of the *npl-4.1* gene is 99% identical to that of the *npl-4.2* gene, the cloned cDNA will target both of these genes. Therefore, the clone is referred to as *npl-4*. The fragment was cloned into the SmaI and XbaI sites of pL4440 (Open Biosystems) and transformed into *E. coli* HT115(DE3) cells.

## Plasmid constructs and generation of transgenic *C. elegans*

For overexpression of *elt-2*, the *elt-2* gene along with its promoter region (1980 bp upstream) was amplified from genomic DNA of N2 animals. The gene, including its stop codon, was cloned in the pPD95.77 plasmid using the restriction sites PstI and SmaI. N2 worms were microinjected with *elt-2p::elt-2* plasmid along with pCFJ90 (*myo-2p::mCherry*) as a coinjection marker to generate the overexpression strain, *jsnEx3 [elt-2p::elt-2+myo-2p::mCherry]*. The *elt-2p::elt-2* plasmid was used at a concentration of 50 ng/µL, while the coinjection marker was used at a concentration of 5 ng/µL.

## RNA interference

RNAi was used to generate loss-of-function phenotypes by feeding worms with *E. coli* strain HT115(DE3) expressing double-stranded RNA homologous to a target *C. elegans* gene. RNAi was carried out as described previously (*Das et al., 2023*). Briefly, *E. coli* HT115(DE3) with the appropriate vectors was

grown in LB broth containing ampicillin (100 μg/mL) at 37°C overnight on a shaker, concentrated 10 times, and plated onto RNAi NGM plates containing 100 μg/mL ampicillin and 3 mM isopropyl β-D-thiogalactoside. The plated bacteria were allowed to grow overnight at 37°C. For synchronization of worms, gravid adults were transferred to RNAi-expressing bacterial lawns and allowed to lay eggs for 2 hr. The gravid adults were removed, and the eggs were incubated at 20°C for 96 hr. For protease and IPR genes' co-RNAi with *ufd-1*, *E. coli* HT115(DE3) with the appropriate vectors were grown separately at 37°C overnight until growth saturation. Then, the RNAi cultures were mixed in a ratio of 1:1, concentrated 10 times, and plated onto RNAi plates, followed by overnight growth at 37°C. We used all the protease and *pals* genes that were upregulated upon *ufd-1* RNAi and were present in the Ahringer RNAi library. For experiments involving *elt-2* RNAi, the worms were grown on the control *gfp* RNAi (GFP) and *ufd-1* RNAi for 48 hr at 20°C to obtain the L4 stage worms. Afterward, the worms grown on *gfp* RNAi were transferred to GFP+*elt-2* RNAi plates, and those grown on *ufd-1* RNAi were transferred to GFP+*ufd-1* and *ufd-1*+*elt-2* RNAi plates. This was followed by the incubation of the worms at 20°C for another 48 hr before transferring to *P. aeruginosa* plates. The *gfp* RNAi *E. coli* HT115(DE3) strain was a kind gift from Scott G Kennedy, Harvard Medical School.

## *C. elegans* killing assay on *P. aeruginosa* PA14

The full-lawn killing assays of *C. elegans* on *P. aeruginosa* PA14 were carried out as described earlier (*Singh and Aballay, 2019a*). Briefly, *P. aeruginosa* PA14 cultures were grown by inoculating individual bacterial colonies into 2 mL of LB broth and growing them for 8–10 hr on a shaker at 37°C. Then, 20 μL of the culture was spread on the complete surface of 3.5-cm-diameter standard slow-killing (SK) plates (modified NGM agar plates [0.35% instead of 0.25% peptone]). The plates were incubated at 37°C for 12–16 hr and then cooled to room temperature for at least 30 min before seeding with synchronized gravid adult hermaphrodite worms. The killing assays were performed at 25°C, and live animals were transferred to fresh plates every 24 hr. Animals were scored at the indicated times and considered dead when they failed to respond to touch. At least three independent experiments were performed for each condition.

## *P. aeruginosa* - GFP colonization assay

The *P. aeruginosa* PA14-GFP colonization assays were carried out as described earlier (*Das et al., 2023*; *Singh and Aballay, 2019b*). Briefly, bacterial cultures were prepared by inoculating individual bacterial colonies into 2 mL of LB broth containing 50 μg/mL kanamycin and growing them for 8–10 hr on a shaker at 37°C. Then, 20 μL of the culture was spread on the complete surface of 3.5-cm-diameter SK plates containing 50 μg/mL kanamycin. The plates were incubated at 37°C for 12–16 hr and then cooled to room temperature for at least 30 min before seeding with gravid adult hermaphrodite worms. The assays were performed at 25°C. At indicated times, the worms were picked under a non-fluorescence stereomicroscope and visualized within 5 min under a fluorescence microscope.

## Quantification of intestinal bacterial loads

The quantification of intestinal *P. aeruginosa* PA14-GFP load was carried by measuring CFUs as described earlier (*Das et al., 2023*). Briefly, *P. aeruginosa* PA14-GFP lawns were prepared as described above. At the indicated times for each experiment, the animals were transferred from *P. aeruginosa*-GFP plates to the center of fresh *E. coli* OP50 plates thrice for 10 min each to eliminate bacteria stuck to their body. Afterward, 10 animals/condition were transferred into 50 μL of PBS containing 0.01% Triton X-100 and ground using glass beads. Serial dilutions of the lysates ($10^1$, $10^2$, $10^3$, $10^4$) were seeded onto LB plates containing 50 μg/mL of kanamycin to select for *P. aeruginosa*-GFP cells and grown overnight at 37°C. Single colonies were counted the next day and represented as the number of bacterial cells or CFUs per animal. Six independent experiments were performed for each condition.

## Pharyngeal pumping assay

For the pharyngeal pumping assay without *P. aeruginosa* PA14 exposure, wild-type N2 animals were grown on appropriate RNAi clones till 1-day-old adults before the measurements. For the pharyngeal pumping assay with *P. aeruginosa* PA14 exposure, wild-type N2 animals were grown on appropriate RNAi clones till 1-day-old adults, followed by exposure to *P. aeruginosa* PA14 for 12 hr at 25°C before

measurements. The number of contractions of the terminal bulb of the pharynx was counted for 30 s per worm. The pumping rates for at least 30 worms were recorded for each condition.

## Measurement of DMP rate

The wild-type N2 animals were synchronized and grown at 20°C on EV, *ufd-1*, and *npl-4* RNAi clones till 1-day-old adults before the measurements. For the DMP assay involving exposure of *C. elegans* to *P. aeruginosa* PA14, wild-type N2 animals were synchronized and grown at 20°C on EV, *ufd-1*, and *npl-4* RNAi clones till 1-day-old adults, followed by exposure to *P. aeruginosa* PA14 for 12 hr at 25°C before measurements. The DMP cycle length was scored by assessing the time between expulsions (which are preceded by posterior and anterior body wall muscle contraction and the contraction of enteric muscles in a normal, regular pattern) (*Thomas, 1990*). The number of expulsion events in 15 min was measured for each worm. The DMP rate was recorded for 9–13 worms/condition.

## *C. elegans* lifespan assays

Lifespan assays were performed as described earlier (*Das et al., 2023*). Briefly, the assays were performed on RNAi plates containing *E. coli* HT115(DE3) with appropriate vectors in the presence of 50 µg/mL of 5-fluorodeoxyuridine (FUdR). Animals were synchronized on RNAi plates without FUdR and incubated at 20°C. At the late L4 larval stage, the animals were transferred onto the corresponding RNAi plates containing 50 µg/mL of FUdR and incubated at 20°C. Animals were scored every day as live, dead, or gone. Animals that failed to display touch-provoked movement were scored as dead. Animals that crawled off the plates were censored. Experimental groups contained more than 60 animals per condition per replicate. Young adult animals were considered day 0 for the lifespan analysis. Three independent experiments were performed.

## RNA isolation, RNA sequencing, and data analysis

RNA isolation was carried out as described earlier (*Singh and Aballay, 2017*). Briefly, animals were synchronized by egg laying. Approximately 35 N2 gravid adult animals were transferred to 10 cm RNAi plates seeded with control empty vector and *ufd-1* RNAi bacteria and allowed to lay eggs for 4 hr. The gravid adults were then removed, and the eggs were allowed to develop at 20°C for 96 hr. The animals were then collected, washed with M9 buffer, and frozen in TRIzol reagent (Life Technologies, Carlsbad, CA, USA). Total RNA was extracted using the RNeasy Plus Universal Kit (QIAGEN, Netherlands). Residual genomic DNA was removed using TURBO DNase (Life Technologies, Carlsbad, CA, USA). Library preparation and sequencing were performed at the Novogene Corporation Inc, USA. The cDNA libraries were sequenced on the HiSeqX sequencing platform using 150 bp paired-end nucleotide reads.

The RNA sequence data were analyzed using the web platform Galaxy (https://usegalaxy.org/). The paired reads were first trimmed using the Trimmomatic tool. The trimmed reads obtained for each sample were mapped to the *C. elegans* genome (WS220) using the aligner STAR. The number of reads mapped to each gene was counted using the *htseq-count* tool. Differential gene expression analysis was then performed using DESeq2. Genes exhibiting at least a twofold change and p-value<0.01 were considered differentially expressed. GO analysis was performed using the DAVID Bioinformatics Database (https://david.ncifcrf.gov/tools.jsp). The overlap of the upregulated genes with previously published datasets was carried out with WormExp v 2.0 (https://wormexp.zoologie.uni-kiel.de/wormexp/) (*Yang et al., 2016b*). The Venn diagrams were obtained using the web tool BioVenn (https://www.biovenn.nl/) (*Hulsen et al., 2008*).

## RNA isolation and qRT-PCR

Animals were synchronized by egg laying. Approximately 40 N2 gravid adults were transferred to 9 cm RNAi plates seeded with *E. coli* HT115 expressing the appropriate vectors and allowed to lay eggs for 4 hr. The adults were then removed, and the eggs were allowed to develop at 20°C for 96 hr. The resulting animals were collected, washed with M9 buffer three times, and frozen in TRIzol reagent (Life Technologies, Carlsbad, CA, USA). Total RNA was extracted using the RNeasy Plus Universal Kit (QIAGEN, Netherlands). qRT-PCR was carried out as described earlier (*Ghosh and Singh, 2024*). Briefly, total RNA was reverse-transcribed with random primers using the PrimeScript 1st strand cDNA Synthesis Kit (TaKaRa) according to the manufacturer's protocols. qRT-PCR was conducted using TB

Green fluorescence (TaKaRa) on a MasterCycler EP Realplex 4 thermal cycler (Eppendorf) in 96-well plate format. Fifteen microliter reactions were analyzed as outlined by the manufacturer (TaKaRa). Relative fold changes of the transcripts were calculated using the comparative $CT$ ($2^{-\Delta\Delta CT}$) method and normalized to pan-actin (*act-1, -3, -4*) as previously described (*Singh and Aballay, 2017*). All samples were run in triplicate (technical replicates) and repeated at least four times (biological replicates).

### Fluorescence imaging

Fluorescence imaging was carried out as described previously (*Gokul and Singh, 2022*; *Ravi, et al., 2023*). Briefly, the animals were anesthetized using an M9 salt solution containing 50 mM sodium azide and mounted onto 2% agarose pads. The animals were then visualized using a Nikon SMZ-1000 fluorescence stereomicroscope. The fluorescence intensity was quantified using ImageJ software.

### Quantification and statistical analysis

The statistical analysis was performed with Prism 8 (GraphPad). All error bars represent mean ± standard deviation (SD). The two-sample t-test was used when needed, and the data were judged to be statistically significant when $p < 0.05$. In the figures, asterisks (*) denote statistical significance as follows: *, $p < 0.05$, **, $p < 0.01$, ***, $p < 0.001$, as compared with the appropriate controls. The Kaplan-Meier method was used to calculate the survival fractions, and statistical significance between survival curves was determined using the log-rank test. All experiments were performed in triplicate.

## Acknowledgements

Some strains used in this study were provided by the Caenorhabditis Genetics Center (CGC), which is funded by the NIH Office of Research Infrastructure Programs (P40 OD010440). This work was supported by the following grants: Ramalingaswami Re-entry Fellowship (Ref. No. BT/RLF/Re-entry/50/2020) and Har-Gobind Khorana-Innovative Young Biotechnologist Fellowship (File No. HRD-17011/2/2023-HRD-DBT) awarded by the Department of Biotechnology, India; STARS grant (File No. MoE-STARS/STARS-2/2023-0116) awarded by the Ministry of Education, India; Research Grant (Ref. No. 37/1741/23/EMR-II) awarded by the Council of Scientific & Industrial Research (CSIR), India; Science and Engineering Research Board (SERB) Startup (Ref. No. SRG/2020/000022) and Core (Ref. No. CRG/2023/001136) Research Grants awarded by DST, India; and IISER Mohali intramural funds. RR was supported by a senior research fellowship from the CSIR, India.

## Additional information

### Funding

| Funder | Grant reference number | Author |
| --- | --- | --- |
| Department of Biotechnology, Ministry of Science and Technology, India | BT/RLF/Re-entry/50/2020 | Jogender Singh |
| Department of Biotechnology, Ministry of Science and Technology, India | HRD-17011/2/2023-HRD-DBT | Jogender Singh |
| Ministry of Education, India | MoE-STARS/STARS-2/2023-0116 | Jogender Singh |
| Council of Scientific and Industrial Research, India | 37/1741/23/EMR-II | Jogender Singh |
| Science and Engineering Research Board | SRG/2020/000022 | Jogender Singh |
| Science and Engineering Research Board | CRG/2023/001136 | Jogender Singh |

| Funder | Grant reference number | Author |
|--------|------------------------|--------|
| CSIR INDIA | | Rajneesh Rao |
| IISER Mohali | | |

The funders had no role in study design, data collection and interpretation, or the decision to submit the work for publication.

## Author contributions
Rajneesh Rao, Conceptualization, Data curation, Formal analysis, Investigation, Methodology; Alejandro Aballay, Conceptualization, Supervision, Visualization; Jogender Singh, Conceptualization, Data curation, Formal analysis, Supervision, Funding acquisition, Validation, Investigation, Visualization, Methodology, Writing - original draft, Writing - review and editing

## Author ORCIDs
Rajneesh Rao ⓘ https://orcid.org/0009-0002-5808-3243
Alejandro Aballay ⓘ https://orcid.org/0000-0002-5975-3352
Jogender Singh ⓘ https://orcid.org/0000-0002-7947-0405

Reviewer #1 (Public review): https://doi.org/10.7554/eLife.94310.3.sa1
Reviewer #2 (Public review): https://doi.org/10.7554/eLife.94310.3.sa2
Author response https://doi.org/10.7554/eLife.94310.3.sa3

# Additional files

## Supplementary files
Supplementary file 1. Upregulated and downregulated genes in *ufd-1* RNA interference (RNAi) versus empty vector (EV) control RNAi N2 animals. Genes exhibiting at least a twofold change and p-value<0.01 were considered differentially expressed.

MDAR checklist

## Data availability
The RNA sequencing data for N2 worms grown on empty vector control and ufd-1 RNAi have been submitted to the public repository, the Sequence Read Archive, with BioProject ID PRJNA1033335. All data generated or analyzed during this study are included in the manuscript and supporting files.

The following dataset was generated:

| Author(s) | Year | Dataset title | Dataset URL | Database and Identifier |
|-----------|------|---------------|-------------|-------------------------|
| Rao R, Aballay A, Singh J | 2023 | *C. elegans* RNA seq data for control and ufd-1 RNAi | https://www.ncbi.nlm.nih.gov/bioproject/PRJNA1033335/ | NCBI BioProject, PRJNA1033335 |

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
