## [Editor Report · eLife Assessment]

In this **valuable** manuscript, Rao and colleagues investigate the UFD-1/NPL-4 complex, which is involved in extracting misfolded proteins in the plasma membrane and the accumulation of pathogenic bacteria in the intestine. Using **convincing** methods, the authors find that knockdown of the *ufd-1* and *npl-4* genes leads to shortened lifespan of the nematode *C. elegans* and reduced accumulation of the bacterial pathogen *P. aeruginosa* in the intestine.

---

## [Referee Report · Reviewer #1 (Public review)]

The authors adequately addressed the concerns I raised in my initial review, which are noted below.

(1) I suggest that the authors choose a different term in their title, abstract and manuscript to describe the phenotypes associated with ufd-1 and npl-4 knockdown other than an "inflammation-like response." Inflammation is a pathological term with four cardinal signs: redness (rubor), swelling (tumor), warmth (calor) and pain (dolor). These are not symptoms known to occur in *C. elegans*. The authors could consider using "inappropriate," "aberrant" or "toxic" immune activation in the title and abstract.

(2) I think it is important to point out in the context of the authors novelty claim in the abstract and manuscript that the toxic effects of inappropriate immune activation in *C. elegans* has been widely catalogued. For example: doi.org/10.1371/journal.ppat.1011120 (2023); doi:10.1186/s12915-016-0320-z (2016).; doi:10.1126/science.1203411 (2011); doi:10.1534/g3.115.025650 (2016). In addition, doi:10.7554/eLife.74206 (2022) previously described a mutation that caused innate immune activation that reduced accumulation of *P. aeruginosa* in the intestine, but also caused animals to have a shortened lifespan.

Thus, I do not think this study reveals the existence of inflammatory-like responses in *C. elegans*, as stated by the authors. Indeed, I think it is important for the authors to remove this novelty claim from their paper and discuss their work in the context of these studies in a paragraph in the introduction.

(3) The authors rely on the use of RNAi of ufd-1 and npl-4 to study their effect on *P. aeruginosa* colonization and pathogen resistance throughout the manuscript. To address the possibility of off-target effects of the RNAi, the authors should consider both (i) showing with qRT-PCR that these genes are indeed targeted during RNAi, and (ii) confirming their phenotypes with an orthologous technique, preferably by studying ufd-1 and npl-4 loss-of-function mutants [both in the wild-type and sek-1(km4) backgrounds]. If mutation of these genes is lethal, the authors could use Auxin Inducible Degron (AID) technology to induce the degradation of these proteins in post-developmental animals.

(4) I am confused about the author's explanation regarding their observation that inhibition of the UFD-1/ NPL-4 complex extends the lifespan of sek-1(km25) animals, but not pmk-1(km25) animals, as SEK-1 is the MAPKK that functions immediately upstream of the p38 MAPK PMK-1 to promote pathogen resistance.

I am also confused why their RNA-seq experiment revealed a signature of intracellular pathogen response genes and not PMK-1 targets, which the authors propose is accounting for toxic immune activation. Activation of which immune response leads to toxicity?

(5) The authors did not test alternative explanations for why UFD-1/ NPL-4 complex inhibition compromises survival during pathogen infection, other than exuberant immune activation. For example, it is possible that inhibition of this proteosome complex shortens lifespan by compromising the general health/ normal physiology of nematodes. Immune responses could be activated as a secondary consequence of this stress, and not be a direct cause of early mortality. Does sek-1(km4) mutant suppress the lifespan shortened lifespan of ufd-1 and npl-4 knockdown? This experiment should also be done with loss-of-function mutants, as noted in point 3.

(6) The conclusion of Figure 6 hinges on an experiment that uses double RNAi to knockdown two genes at the same time (Fig. 6D and 6G), an approach that is inherently fraught in *C. elegans* biology owing to the likelihood that the efficiency of RNAi-mediated gene knockdown is compromised and may account for the observed phenotypes. The proper control for double RNAi is not empty vector + ufd-1(RNAi), but rather gfp(RNAi) + ufd-1(RNAi), as the introduction of a second hairpin RNA is what may compromise knockdown efficiency. In this context, it is important to confirm that knockdown of both genes occurs as expected (with qRT-PCR) and to confirm this phenotype using available elt-2 loss-of-function mutants.

(7) A supplementary table with the source data for at least three replications (mean lifespan, n, statistical comparison) for each pathogenesis assay should be included in this manuscript.

Comments on revisions:

The authors adequately addressed the concerns I raised.

---

## [Referee Report · Reviewer #2 (Public review)]

Summary:

The authors aimed to uncover what role, if any, the UFD1/NPL4 complex might play in innate immune responses of the nematode *C. elegans*. The authors find that loss of the complex renders animals more sensitive to both pathogenic and non-pathogenic bacteria. However, there appears to be a complex interplay with known innate immune pathways since loss of UFD1/NPL4 actually results in increased survival of animals lacking the canonical innate immune pathways.

Strengths:

The authors perform robust genetic analysis to exclude and include possible mechanisms by which the UFD1/NPL4 pathway acts in the innate immune response.

Weaknesses:

The argument that the loss of the UFD1/NPL4 complex triggers a response that mimics that of an intracellular pathogen is not thoroughly investigated. Additionally, the finding of a role of the GATA transcription factor, ELT-2, in this response is suggestive, but experiments showing sufficiency in the context of loss of the UFD1/NPL4 complex need to be explored.

Comments on revisions:

The authors have performed several control experiments for their RNAi based experiments and also tested the requirement for xbp-1s in their paradigm. The findings and their interpretations are acceptable.

---

## [Author Response]

The following is the authors’ response to the original reviews.

**Reviewer #1 (Public Review):**
(1) I suggest that the author's choose a different term in their title, abstract and manuscript to describe the phenotypes associated with ufd-1 and npl-4 knockdown other than an "inflammation-like response." Inflammation is a pathological term with four cardinal signs: redness (rubor), swelling (tumor), warmth (calor) and pain (dolor). These are not symptoms know to occur in *C. elegans*. The authors could consider using "tolerance" instead, as this term may better describe their findings.

We have changed “inflammation-like response” to “aberrant immune response” throughout the manuscript.

(2) It would help the reader to better understand the novelty of the findings in this study if the authors include a paragraph in their introduction to put their results in context of the published literature that has examined the relationship between immune activation and nematode health and survival. In particular, I suggest that the authors discuss doi:10.7554/eLife.74206 (2022), a study that charcterized a similar observation to what the authors are reporting. This study found that low cholesterol reduces pathogen tolerance and host survival during pathogen infection. Cholesterol scarcity increases p38 PMK-1 phosphorylation, priming immune effector induction in a manner that reduces pathogen accumulation in the intestine during a subsequent infection. I also suggest that the authors highlight in this introductory paragraph that the toxic effects of inappropriate immune activation in *C. elegans* has been widely catalogued. For example: doi.org/10.1371/journal.ppat.1011120 (2023); doi:10.1186/s12915-016-0320-z (2016).; doi:10.1126/science.1203411 (2011); doi:10.1534/g3.115.025650 (2016).In this context, the authors could consider re-wording their novelty claim in the abstract and introduction to take into account this previous body of work.

We have added a paragraph to the Discussion section to place our findings in the context of previous research. The revised manuscript now includes the following text (page 11, lines 336–344): “Previous studies have shown that hyperactivation of immune pathways can negatively affect organismal development. For example, sustained activation of the p38 MAPK pathway impairs development in *C. elegans* (Cheesman et al., 2016; Kim et al., 2016), and excessive activation of the IPR also leads to developmental defects (Lažetić et al., 2023). Similar to our current study, recent work has demonstrated that heightened immune responses can reduce gut pathogen load while paradoxically decreasing host survival during infection (Ghosh and Singh, 2024; Peterson et al., 2022). However, our study uniquely shows that while such heightened immune responses are detrimental to immunocompetent animals, they can be beneficial in the context of immunodeficiency.”

(3) The authors rely on the use of RNAi of ufd-1 and npl-4 to study their effect on *P. aeruginosa* colonization and pathogen resistance throughout the manuscript. To address the possibility of off-target effects of the RNAi, the authors should consider both (i) showing with qRT-PCR that these genes are indeed targeted during RNAi, and (ii) confirming their phenotypes with an orthologous technique, preferably by studying ufd-1 and npl-4 loss-offunction mutants [both in the wild-type and sek-1(km4) backgrounds]. If mutation of these genes is lethal, the authors could use Auxin Inducible Degron (AID) technology to induce the degradation of these proteins in post-developmental animals.

We attempted several protocols of CRISPR in our laboratory to generate ufd-1 loss-of-function mutants; however, these efforts were unsuccessful. While this does not rule out the possibility of generating ufd-1 mutants, the failure is likely due to technical limitations on our part rather than an inherent inability to disrupt the gene. Nevertheless, to confirm the specificity of our RNAi-based approach, we quantified ufd-1 and npl-4 mRNA levels following RNAi treatment and found that each gene was specifically and effectively downregulated by its respective RNAi.

Importantly, ufd-1 and npl-4 RNA sequences do not share significant homology, yet knockdown of either gene results in nearly identical phenotypes, including reduced survival on *P. aeruginosa*, diminished intestinal colonization, and shortened lifespan. These consistent outcomes strongly support the conclusion that the phenotypes are attributable to the disruption of the functional UFD-1-NPL-4 complex. We have added these results in the revised manuscript (pages 4-5, lines 114-125): “To confirm the specificity of the RNAi knockdowns and rule out potential off-target effects, we examined transcript levels of ufd-1 and npl-4 following RNAi treatment. RNAi against ufd-1 significantly reduced ufd-1 mRNA levels without reducing npl-4 expression, while npl-4 RNAi specifically downregulated npl-4 transcripts with no impact on ufd-1 mRNA levels (Figure 1—figure supplement 1A and B). Additionally, alignment of ufd-1 and npl-4 mRNA sequences against the *C. elegans* transcriptome revealed no significant similarity to other genes, supporting the specificity of the RNAi constructs. Moreover, the ufd-1 and npl-4 RNA sequences do not share significant sequence similarity. Therefore, the highly similar phenotypes observed in ufd-1 and npl-4 knockdown animals, including shortened lifespan, reduced survival on *P. aeruginosa*, and decreased intestinal colonization with *P. aeruginosa*, strongly suggest that these outcomes result from the disruption of the functional UFD-1-NPL-4 complex.”

(4) I am confused about the authors explanation regarding their observation that inhibition of the UFD-1/ NPL-4 complex extends the lifespan of sek-1(km25) animals, but not pmk-1(km25) animals, as SEK-1 is the MAPKK that functions immediately upstream of the p38 MAPK PMK-1 to promote pathogen resistance.I am also confused why their RNA-seq experiment revealed a signature of intracellular pathogen response genes and not PMK-1 targets, which the authors propose is accounting for toxic immune activation. Activation of which immune response leads to toxicity?

We consistently observe that sek-1(km4) mutants are more sensitive to *P. aeruginosa* infection than pmk-1(km25) mutants, a finding also reported in previous studies (for example, PMID: 33658510). Given that SEK-1 functions upstream of PMK-1 in the MAPK signaling cascade, it is plausible that SEK-1 also regulates additional MAP kinases, such as PMK-2 (PMID: 25671546), which could contribute to the enhanced susceptibility observed in sek-1 mutants.

Our results show that inhibition of the UFD-1-NPL-4 complex improves survival specifically in severely immunocompromised animals, such as sek-1(km4) mutants, but not in pmk1(km25) mutants. To further validate this, we generated the double mutant dbl-1(nk3);pmk1(km25), which exhibits reduced survival on *P. aeruginosa* compared to either single mutant.

Notably, inhibition of the UFD-1-NPL-4 complex also enhances survival in the dbl1(nk3);pmk-1(km25) background, reinforcing the observation that this response is specific to severely compromised immune states.

We would also like to clarify that the observed phenotypes are independent of the SEK1/PMK-1 pathway, as shown in Figure 3A-3C, Figure 3—figure supplement 1, and Figure 4A-4C. The IPR seems to play a role in the observed phenotypes, as inhibition of some of the protease and pals genes (IPR genes) leads to increased *P. aeruginosa* colonization in ufd-1 knockdown animals (Figure 6—figure supplement 1). The other immune response pathway that leads to the observed phenotypes is ELT-2, as explained in Figure 6. Finally, we have included in the revised manuscript a note that, in addition, as-yet unidentified pathways are also likely contributing to the phenotypes triggered by disruption of the UFD-1-NPL-4 complex.

(5) The authors did not test alternative explanations for why UFD-1/ NPL-4 complex inhibition compromises survival during pathogen infection, other than exuberant immune activation. For example, it is possible that inhibition of this proteosome complex shortens lifespan by compromising the general health/ normal physiology of nematodes. Immune responses could be activated as a secondary consequence of this stress, and not be a direct cause of early morality. Does sek-1(km4) mutant suppress the lifespan shortened lifespan of ufd-1 and npl-4 knockdown? This experiment should also be done with loss-offunction mutants, as noted in point 3.

We have already included this data in Figure 4D, where we observed that ufd-1 and npl-4 knockdown reduce the lifespan of sek-1(km4) animals. It is possible that immune activation is a secondary consequence of cellular stress induced by inhibition of the UFD-1NPL-4 complex. However, our data strongly suggest that the observed phenotypes, including reduced gut pathogen load and decreased survival on the pathogen, are due to the aberrant immune response activated by the inhibition of the UFD-1-NPL-4 complex. Evidence from sek-1(km4) mutants particularly underscores the role of this dysregulated immune activation. While this aberrant immune response is detrimental to wild-type animals under pathogenic conditions, it appears to be beneficial in severely immunocompromised backgrounds. Specifically, in sek-1(km4) mutants, inhibition of the UFD-1-NPL-4 complex enhances survival during *P. aeruginosa* infection (Figure 4A). However, under non-infectious conditions, where sek-1(km4) mutants exhibit a normal lifespan, the same immune activation becomes harmful (Figure 4D). Together, these findings demonstrate that the aberrant immune response induced by UFD-1–NPL-4 inhibition is context-dependent: it is advantageous only for immunocompromised animals under infection, but deleterious to healthy animals under infection and to both healthy and immunocompromised animals under non-infectious conditions.

(6) The conclusion of Figure 6 hinges on an experiments that uses double RNAi to knockdown two genes at the same time (Fig. 6D and 6G), an approach that is inherently fraught in *C. elegans* biology owing the likelihood that the efficiency of RNAi-mediated gene knockdown is compromised and may account for the observed phenotypes. The proper control for double RNAi is not empty vector + ufd-1(RNAi), but rather gfp(RNAi) + ufd1(RNAi), as the introduction of a second hairpin RNA is what may compromise knockdown efficiency. In this context, it is important to confirm that knockdown of both genes occurs as expected (with qRT-PCR) and to confirm this phenotype using available elt-2 loss-of-function mutants.

We thank the reviewer for this helpful suggestion. We have repeated all double

RNAi experiments using gfp RNAi as a control instead of the empty vector (Figure 6 and Figure 6—figure supplement 1). Additionally, we assessed the efficiency of gene knockdown in the double RNAi conditions (Figure 6—figure supplement 2) and found that RNAi efficacy was not compromised by the double RNAi treatment.

(7) A supplementary table with the source data for at least three replications (mean lifespan, n, statistical comparison) for each pathogenesis assay should be included in this manuscript.

The source data is provided for all the data presented in the manuscript.

**Reviewer #2 (Public Review):**
Summary:The authors aimed to uncover what role, if any, the UFD1/NPL4 complex might play in the innate immune responses of the nematode *C. elegans*. The authors find that loss of the complex renders animals more sensitive to both pathogenic and non-pathogenic bacteria. However, there appears to be a complex interplay with known innate immune pathways since the loss of UFD1/NPL4 actually results in increased survival of animals lacking the canonical innate immune pathways.

We thank the reviewer for providing an excellent summary of our work.

Strengths:The authors perform robust genetic analysis to exclude and include possible mechanisms by which the UFD1/NPL4 pathway acts in the innate immune response.

We thank the reviewer for highlighting the strengths of our work.

Weaknesses:The argument that the loss of the UFD1/NPL4 complex triggers a response that mimics that of an intracellular pathogen has not been thoroughly investigated. Additionally, the finding of a role of the GATA transcription factor, ELT-2, in this response is suggestive, but experiments showing sufficiency in the context of loss of the UFD1/NPL4 complex need to be explored.

We have investigated the role of IPR genes in the phenotypes observed upon ufd1 knockdown (Figure 6—figure supplement 1), and our results suggest that the IPR may contribute, at least in part, to the phenotypic outcomes of ufd-1 RNAi. In the Discussion section (pages 11–12, lines 345–356), we have included a detailed discussion on the possible mechanisms underlying IPR activation upon inhibition of the UFD-1–NPL-4 complex. We agree that the interaction between the UFD-1–NPL-4 complex and the IPR is intriguing and warrants further investigation. However, we believe that an in-depth exploration of this interaction lies beyond the scope of the current study.

We have incorporated new data on ELT-2 overexpression in the revised manuscript. Overexpression of ELT-2 partially phenocopies the effects of ufd-1 knockdown, supporting the idea that other pathways likely contribute to the full spectrum of phenotypes observed upon UFD-1-NPL-4 complex inhibition. The revised manuscript reads (page 10, lines 311319): “To determine whether ELT-2 activation alone is sufficient to recapitulate the phenotypes observed upon UFD-1-NPL-4 complex inhibition, we analyzed animals overexpressing ELT-2. Similar to ufd-1 knockdown, ELT-2 overexpression led to a significant reduction in the colonization of the gut by *P. aeruginosa* (Figure 6—figure supplement 3A and 3B). However, overexpression of ELT-2 did not alter the survival of worms on *P. aeruginosa* (Figure 6—figure supplement 3C). Taken together, these findings suggest that the phenotypes triggered by disruption of the UFD-1-NPL-4 complex are partially mediated by ELT-2. However, additional pathways, yet to be identified, likely cooperate with ELT-2 to regulate both pathogen resistance and host survival.”

**Reviewer #1 (Recommendations For The Authors):**
The authors could consider avoiding the use of descriptors (e.g., "drastic") when presenting their data.

We have removed the descriptors.

**Reviewer #2 (Recommendations For The Authors):**
What happens with overexpression of ELT2?

Overexpression of ELT-2 partially recapitulates the phenotypes of ufd-1 knockdowns, indicating that additional pathways are likely involved in controlling the phenotypes observed upon inhibition of the UFD-1-NPL-4 complex. The revised manuscript reads (page 10, lines 311-319): “To determine whether ELT-2 activation alone is sufficient to recapitulate the phenotypes observed upon UFD-1-NPL-4 complex inhibition, we analyzed animals overexpressing ELT-2. Similar to ufd-1 knockdown, ELT-2 overexpression led to a significant reduction in the colonization of the gut by *P. aeruginosa* (Figure 6—figure supplement 3A and 3B). However, overexpression of ELT-2 did not alter the survival of worms on *P. aeruginosa* (Figure 6—figure supplement 3C). Taken together, these findings suggest that the phenotypes triggered by disruption of the UFD-1-NPL-4 complex are partially mediated by ELT-2. However, additional pathways, yet to be identified, likely cooperate with ELT-2 to regulate both pathogen resistance and host survival.”

The data with xbp-1 loss of function is very different than that of pek1 and atf-6. Does loss of ufd1/npl4 suppress the increased pathogen survival of xbp-1s overexpressing animals?

We have examined worms overexpressing XBP-1s and found that overexpression of XBP-1s does not rescue the phenotypes caused by ufd-1 knockdown. The revised manuscript reads (page 6, lines 167-174): “To further examine the role of XBP-1 in this context, we assessed the effect of ufd-1 knockdown in animals neuronally overexpressing the constitutively active spliced form of XBP-1 (XBP-1s), which has been previously associated with enhanced longevity (Taylor and Dillin, 2013). Knockdown of ufd-1 resulted in the reduced survival of XBP-1s-overexpressing animals on *P. aeruginosa*, despite a concurrent decrease in bacterial colonization of the gut (Figure 2—figure supplement 1A-C). This indicated that the XBP-1 pathway was not required for the reduced *P. aeruginosa* colonization of ufd-1 knockdown animals.”

Lastly, while the pathogen burden is reduced in ufd1/npl4 loss and pumping rates are marginally affected, have you checked defecation rates? Could they be increased?

We thank the reviewer for this valuable suggestion. We measured defecation rates following ufd-1 and npl-4 knockdown and, unexpectedly, found that inhibition of ufd-1/npl-4 leads to a reduction in defecation frequency. These findings clearly indicate that altered defecation cannot explain the observed decrease in gut colonization. The revised manuscript reads (page 5, lines 138-148): “The clearance of intestinal contents through the defecation motor program (DMP) is known to influence gut colonization by *P. aeruginosa* in *C. elegans* (Das et al., 2023). It is therefore conceivable that knockdown of the UFD-1-NPL-4 complex might increase defecation frequency, thereby promoting the physical expulsion of bacteria and resulting in reduced gut colonization. To test this possibility, we measured DMP rates in animals subjected to ufd-1 and npl-4 RNAi. Contrary to this hypothesis, both ufd-1 and npl-4 knockdown animals exhibited a significant reduction in defecation frequency compared to control RNAi-treated animals (Figure 1—figure supplement 2C). This reduction in DMP rate persisted even after 12 hours of exposure to *P. aeruginosa* (Figure 1—figure supplement 2D). Thus, the change in the DMP rate in ufd-1 and npl-4 knockdown animals is unlikely to be the reason for the reduced gut colonization by *P. aeruginosa*.”

In summary, we would like to thank the reviewers again for providing constructive and thoughtful feedback. We believe we have fully addressed all the concerns of the reviewers by carrying out several new experiments and modifying the text. The manuscript has undergone substantial revision and has thereby improved significantly. We do hope that the evidence in support of the conclusions is found to be complete in the revised manuscript.